# High-throughput 3D engineered paediatric tumour models for precision medicine

MoonSun Jung[1,2,14], Valentina Poltavets[1,3,4,14], Joanna N Skhinas[1,4], Gabor Tax [1,3], Alvin Kamili [1,3], Jinhan Xie[1,3], Sarah Ghamrawi[1,5], Philipp Graber [1,3,4], Jie Mao [1], Marie Wong-Erasmus[1], Louise Cui [1], Kathleen Kimpton [1], Pooja Venkat[1], Chelsea Mayoh [1,3], Angela Lin [1], Emmy D G Fleuren[1,3], Ashleigh M Fordham[1], Zara Barger[1], John Grady [6], David M Thomas[7], Eric Y Du[8], Nicole S Graf [9,10], Mark J Cowley [1,3], Andrew J Gifford [1,3,11], Jamie I Fletcher [1,3], Loretta M S Lau[1,3,12], M Emmy M Dolman[1,3,13], J Justin Gooding [4,8✉] & Maria Kavallaris [1,3,4✉]

## Abstract

Precision medicine for paediatric and adult cancers that incorporates drug sensitivity profiling can identify effective therapies for individual patients. However, obtaining adequate biopsy samples for high-throughput (HTP) screening remains challenging, with tumours needing to be expanded in culture or patient-derived xenografts, this is time-consuming and often unsuccessful. Herein, we have developed paediatric patient-derived tumour models using an engineered extracellular matrix (ECM) tissue mimic hydrogel system and HTP 3D bioprinting. Gene expression analysis from a neuroblastoma and sarcoma paediatric patient cohort identified key components of the ECM in these tumour types. Engineered hydrogels with ECM-mimic peptides were used to bioprint and create patient-specific tumouroids using patient-derived cells from xenograft models, and the approach was further confirmed on direct patient tumour samples. Bioprinted tumouroids from the PDX models recapitulated the genetic and phenotypic characteristics of the original tumours and retained tumourigenicity. HTP drug screening of these models identified individualised drug sensitivities. Our approach offers a timely and clinically relevant technology platform for precision medicine in paediatric cancers, potentially transforming preclinical testing across multiple cancer types.

**Keywords** Paediatric Cancer; Preclinical Models; Tumouroid Models; 3D Bioprinting; High-throughput (HTP) Drug Screening
**Subject Categories** Biotechnology & Synthetic Biology; Cancer

## Introduction

Cancer remains the leading cause of disease-related death in children in developed countries (Steliarova-Foucher et al, 2017). High-risk cancer makes up 30% of all cases and prognosis is poor despite patients receiving intensive treatments that cause significant side effects and dose-limiting toxicities (Smith et al, 2010). Neuroblastoma is the most common extracranial solid tumour, with an overall survival rate of ~60% for high-risk disease (Irwin et al, 2021). Sarcomas are heterogeneous mesenchymal tumours arising from bone or soft tissue, with patients having metastatic disease showing overall survival rates of 30% for Ewing sarcoma (Gaspar et al, 2015) and 45% for osteosarcoma (Reed et al, 2023), respectively. Neuroblastoma and sarcoma patients with high-risk disease have limited therapeutic options. Moreover, many of the therapies given to treat patients with recurrent and drug-resistant disease are highly toxic, which means that children can be exposed to damaging and ineffective therapies. Indeed, survivors have a high likelihood of experiencing life-long health issues (Suh et al, 2020).

With the advancement of sequencing technologies, molecular-based precision medicine has gained momentum in cancer management to identify actionable mutations and associated treatments for paediatric patients (Berlanga et al, 2022; Church et al, 2022; Peterziel et al, 2022; Wong et al, 2020). However, genomic testing alone does not benefit all patients, since ~30% of paediatric patients do not have actionable molecular alterations

[1]Children's Cancer Institute, Lowy Cancer Research Centre, UNSW Sydney, Sydney, NSW 2052, Australia. [2]NHMRC Clinical Trials Centre, Faculty of Medicine and Health, The University of Sydney, Sydney, NSW 2050, Australia. [3]School of Clinical Medicine, UNSW Medicine & Health, UNSW Sydney, Sydney, NSW 2052, Australia. [4]Australian Centre for NanoMedicine, UNSW Sydney, Sydney, NSW 2052, Australia. [5]Melanoma Institute Australia, The University of Sydney, Sydney, NSW 2050, Australia. [6]The Garvan Institute of Medical Research and The Kinghorn Cancer Centre, Darlinghurst, NSW 2010, Australia. [7]The Centre for Molecular Oncology, School of Biomedical Sciences, UNSW Medicine & Health, UNSW Sydney, Sydney, NSW 2052, Australia. [8]School of Chemistry, UNSW, Sydney, NSW 2052, Australia. [9]Histopathology Dept, The Children's Hospital at Westmead, SCHN, Westmead, NSW 2145, Australia. [10]University of Sydney, CHW Clinical School, Westmead, NSW 2145, Australia. [11]Anatomical Pathology, NSW Health Pathology, Prince of Wales Hospital, Randwick, NSW 2031, Australia. [12]Kids Cancer Centre, Sydney Children's Hospital, Sydney, NSW 2031, Australia. [13]Princess Maxima Center for Paediatric Oncology, Utrecht 3584 CS, The Netherlands. [14]These authors contributed equally: MoonSun Jung, Valentina Poltavets. ✉E-mail: justin.gooding@unsw.edu.au; m.kavallaris@ccia.unsw.edu.au

(Lau et al, 2024) and even then, targeting an actionable alteration with a specific drug results in objective clinical response in the minority of patients (Berlanga et al, 2022; Lau et al, 2024). Tumour drug sensitivity profiling (DSP) holds promise to support clinical treatment decision-making in precision medicine and expand therapeutic options for cancer patients (Acanda De La Rocha et al, 2024; Mayoh et al, 2023; Peterziel et al, 2022). In precision medicine platforms, tumour biopsies often lack sufficient material for use beyond routine diagnostic histopathology and genomic analysis, precluding comprehensive DSP. Subsequently, cell expansion is often required via primary cell culture or the development of patient-derived xenograft models (PDXs) (Lau et al, 2022; Napoli et al, 2022; Pauli et al, 2017). Among the most common challenges with the expansion of primary material are that not all samples are able to grow in culture or successfully engraft in vivo. Furthermore, time required for in vivo expansion of patient material using PDXs is highly variable (up to 12 months) (He et al, 2021; Mayoh et al, 2023; Peterziel et al, 2022; Xie et al, 2021).

Ex vivo tumour organoid models—cancer cells grown in three-dimensional (3D) structures—are increasingly being used to test drug responses of both adult (Lee et al, 2018; Weeber et al, 2017) and paediatric (Calandrini et al, 2020; Calandrini et al, 2021) patient tumours. Tumour organoid models that are representative of the patient's tumour can be robustly expanded, reflect tumour heterogeneity, and are amenable to rapid DSP and mechanistic studies, which have the potential to be transformative for precision medicine. However, the ability to grow tumour cells as tumouroids directly from biopsies is highly variable and cancer type dependent (Lampis et al, 2024; Pauli et al, 2017). For example, paediatric osteosarcomas are challenging to grow ex vivo, often requiring either ex vivo expansion or in vivo engraftment to obtain cell numbers required for comprehensive personalised drug screening (Peterziel et al, 2022). Recent studies have had some success with short-term culturing of osteosarcomas using Matrigel (Al Shihabi et al, 2024). Furthermore, there is a lack of high-throughput (HTP) platforms that efficiently support ex vivo growth of patient tumour cells and are also compatible with HTP drug screening (Corallo et al, 2020; Gaebler et al, 2017). Moreover, many ex vivo models of cancers lack consideration of the extracellular matrix (ECM), which can impact cellular signalling and response to therapy (Chaudhuri et al, 2020; Curvello et al, 2023). 3D models that attempt to incorporate the ECM are often grown in an animal-derived matrix such as Matrigel or collagen, which suffer batch-to-batch variability that impacts cellular cues and stiffness (Below et al, 2022; Law et al, 2021). Mimicking relevant physiological components of the tissue extracellular environment, including ECM protein components and matrix stiffness, will be an important advance in precision medicine.

We recently developed a 3D bioprinting platform whereby cultured cells are embedded in synthetic hydrogels with tunable properties that can mimic a range of ECM components and spatial growth constraints of a tumour (Utama et al, 2020; Utama et al, 2021). This platform allows rapid and reproducible encapsulation of cells with high viability and enables the expansion of a range of cell lines and types in ECM-mimic hydrogels (Jung et al, 2022). A key question was whether this approach could be developed to support the growth and expansion of patient tumour cells for precision medicine.

Here, we investigated the potential of HTP 3D bioprinting and this ECM-mimic hydrogel system to establish high-risk neuroblastoma and sarcoma patient-derived tumouroids that maintain the genomic and phenotypic characteristics of the original patient tumours and incorporate these tumouroids into HTP drug testing for precision cancer medicine. In this proof-of-concept study using cells from patient-derived xenograft models and primary tumour cells, we address some key limitations in precision medicine, namely the ability to maintain and expand difficult to grow freshly isolated patient-derived cells in a HTP fashion and conduct robust HTP drug screening in an environment that mimics tumour growth in a timely manner. Our approach holds enormous potential to identify drug sensitivities in cancers and expand therapeutic options for high-risk cancer patients.

# Results

## Defining the extracellular matrix environment of high-risk neuroblastoma and sarcoma tumours

To investigate the extracellular environment conditions for 3D culture of high-risk neuroblastoma and sarcoma patient-derived cells, we analysed tumour-specific expression patterns of ECM genes using recently published methodology (Izzi et al, 2020) and ECM Organisation gene set (Rouillard et al, 2016). RNA sequencing (RNA-seq) data for high-risk paediatric tumours available through the Australian ZERO Childhood Cancer Precision Medicine Program (ZERO) (Wong et al, 2020) was used to perform bioinformatic analysis of the ECM Organisation gene set ($n = 265$) in a cohort of neuroblastoma (NBL) and sarcomas, comprising Ewing sarcoma (EWS), osteosarcoma (OST) and rhabdomyosarcoma (RMS) ($n = 146$) (Fig. 1A). The top 30 of 265 structural ECM genes and their regulators highly expressed across the four cancer types were identified (Fig. 1B). Many of these are translated to collagens (Ricard-Blum, 2011), the most abundant proteins in the human body and fibronectin (Dalton and Lemmon, 2021), abundant in highly organised structures in both interstitial matrix and basement membrane. Further analysis of the core matrisome genes (encoding structural components of the ECM) (Apte and Naba, 2023; Petrov et al, 2023) confirmed significant upregulation of fibronectin 1 (*FN1*) (Fig. 1C) and collagen I (*COL1A1*) (Fig. 1D) relative to other genes in the cohort. Thus, our analysis identified key genes that encode for structural components within childhood neuroblastoma and sarcoma tumours. This knowledge was incorporated in determining the 3D extracellular mimic environment to support the growth of neuroblastoma and sarcoma patient-derived cells. While gene expression analysis provides valuable insights into ECM gene expression, it may not directly reflect protein abundance or post-translational modifications that mediate ECM function. Many ECM proteins undergo complex processing and structural assembly that cannot be inferred from transcriptomic data alone. Further studies are required to address ECM protein expression in patient tumour samples.

## Tunable ECM-mimic hydrogels support paediatric tumour growth

To grow and expand patient-derived samples, current methods that rely on cell culture or establishment of patient-derived xenografts have limited success (Mayoh et al, 2023). For the proof-of-concept phase of this study, we have obtained eight well-characterised

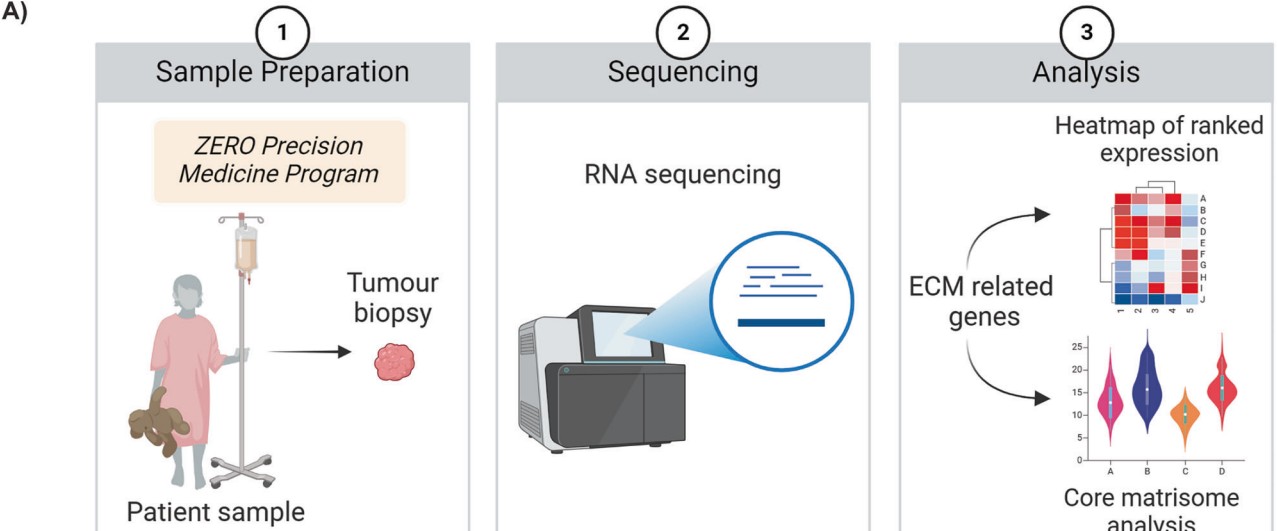

**Figure 1. Defining the extracellular matrix environment of high-risk neuroblastoma and sarcoma tumours.**

(A) Workflow schematic of ECM-related gene analysis from RNA sequencing results. (B) Heatmap of the top and bottom 30 genes ranked by expression levels out of ECM-relevant genes ($n = 265$) identified in neuroblastoma and sarcoma patient samples. Gene transcript per million (TPM) values are represented by colour from red (high Z-score, high expression) to blue (low Z-score, low expression). Diagnosis EWS (Ewing sarcoma $N = 40$); NBL (neuroblastoma $N = 41$); OST (osteosarcoma $N = 25$); RMS (Rhabdomyosarcoma $N = 40$). (C, D) Violin plots of fibronectin (*FN1*) and collagen type I alpha 1 (*COL1A1*) gene expression compared to the remaining core matrisome ECM genes ($n = 79$) in the cohort. Expression levels are log transcript per million (TPM) $+ 1$. Data are presented as median $\pm$IQR, ***$P < 0.001$. Exact $P$ values are $P = 6.39\text{e-}88$ (*FN1*) and $P = 8.15\text{e-}79$ (*COL1A1*) (Welch's $t$ test).

patient-derived xenograft (PDX) models representing high-risk neuroblastomas and sarcomas (Table EV1), that exhibited variable success in generating cell cultures (Table EV2) and length of time to generate patient-derived xenografts (Table EV3). To systematically assess the mechanical and biological requirements for optimal cellular growth across tumour types, we tested the capacity of distinct ECM-mimic hydrogel combinations to support tumour growth in situ. To create a tumour-like extracellular matrix (ECM) for high-risk neuroblastoma and sarcoma, we selected hydrogels with two different stiffness levels: 1.1 kPa and 3 kPa. These stiffness levels closely mimic the lung (Hinz, 2012; Polio et al, 2018) (~0.8 kPa) and liver (Clarke et al, 2011; Yeh et al, 2002) (~3 kPa) tissues, which are frequent metastatic sites for sarcomas (Amankwah et al, 2013) and neuroblastomas (DuBois et al, 1999), respectively.

We prioritised the selection of fibronectin and collagen I based on the top matrisome genes identified in neuroblastoma and sarcoma patient samples (Fig. 1B). Furthermore, we expanded our functional peptide selection with the inclusion of laminin in the hydrogels. Laminin is indispensable for integrin-mediated cell adhesion and a major component of both the basement membrane and Matrigel (Khalilgharibi and Mao, 2021). The hydrogels were engineered to incorporate an integrin-binding peptide of fibronectin (RGD) (Pierschbacher and Ruoslahti, 1984), a collagen I sequence (GFOGER) (Knight et al, 2000), and a laminin peptide fragment (DYIGSR) (Iwamoto et al, 1987). We tested a total of six hydrogel conditions incorporating either fibronectin (FN) alone, a combination of fibronectin and collagen I (FN + CN), or a tripeptide combination of fibronectin, collagen I and laminin (FN + CN + LN), at a stiffness of either 1.1 kPa or 3 kPa.

Patient-derived cells isolated from PDX tumours were bioprinted in the six hydrogel conditions and their proliferative capacity measured by a resazurin-based metabolic assay over a 14-day period. Post-bioprinting, neuroblastoma (Fig. 2Ai), Ewing sarcoma (Fig. 2Bi) and osteosarcoma (Fig. 2Ci) cells in the tripeptide hydrogel (FN + CN + LN) exhibited similar growth dynamics, characterised by consistent proliferative activity up to day 14. Cells remained highly viable for 14 days of culture in both the 1.1 kPa (Fig. 2A–Cii) and 3 kPa (Fig. EV2A–C) tripeptide hydrogel. In addition, we observed similar growth dynamics in one peptide (FN) and two peptide (FN + CN) hydrogels at 1.1 kPa and 3 kPa stiffness (Fig. EV1A–C).

Our analysis examined the influence of hydrogel stiffness and incorporation of functional peptides on cellular growth. Collectively, our data suggest that neither the stiffness, nor specific peptide combinations had a major impact on cellular proliferation across all disease types tested (Figs. 2, EV1 and EV2). However, given the pivotal roles of cell adhesion peptides and the basement membrane for cellular function, along with our findings in the

paediatric tumour ECM-associated genes (Fig. 1C,D), we strategically selected the tripeptide combination of fibronectin, collagen I and laminin-mimicking peptides at 1.1 kPa stiffness for subsequent experiments. This one type of hydrogel was selected for subsequent experiments as it supports the cellular growth of high-risk paediatric tumour cells across multiple cancer types and specifically given that many samples in our cohort were from relapsed metastatic lesions to the lung, the 1.1 kPa stiffness was most appropriate.

## 3D bioprinted tumouroids retain primary molecular characteristics

Having identified the ECM-mimic hydrogel condition supportive of high-risk paediatric neuroblastoma and sarcoma cell growth ex vivo, we systematically investigated the similarity between the 3D bioprinted tumour models and the original tumours. PDX cells were bioprinted, cultured for 7–14 days and retrieved from the hydrogels for targeted capture sequencing of 523 pan-cancer genes (Illumina TruSight Oncology 500) (Fig. 3A). We compared key molecular characteristics between 3D bioprinted PDX tumouroids and the original patient samples previously identified by whole genome sequencing (WGS) and RNA-seq by ZERO (Wong et al, 2020).

Overall, we observed that the molecular drivers of bioprinted PDX cells recapitulated those found in the original patient samples, although the clonal composition of some samples varied slightly. High-risk neuroblastoma bioprinted tumouroids retained disease-specific driver alterations such as *MYCN* (zccs373, zccs154) and *ALK* amplification (zccs154) (Fig. 3B; Table EV4). Ewing sarcoma tumouroids preserved characteristic *EWSR1* rearrangements with erythroblast transformation-specific (ETS) family of transcription factors such as *FLI1*, *ERG* and *ETV1*(Jinawath et al, 2010). Moreover, the Ewing sarcoma sample zccs207 retained *STAG2* (p.Leu1132Ter) nonsense variant, *TERT* (c.-57A > C, 5' UTR) noncoding single-nucleotide variant and *TP53* (p.His193Tyr) missense variant (Fig. 3B; Table EV4). Sample zccs59 preserved *ARID1A* (p.Arg1287LysfsTer11) nonsense variant and *SMARCA4* (p.Arg1157Trp) missense variant. Overall, the tumour-specific subclones were preserved in the bioprinted tumouroids. We observed some clonal enrichment where variant allele frequency (VAF) was higher in zccs59 *(SMARCA4)* and zccs207 *(PIK3CA)* compared to original patient samples (Table EV4). In the case of the Ewing sarcoma zccs227 bioprinted sample, targeted RNA sequencing did not identify the *EWSR1-ERG* fusion. However, subsequent PCR analysis confirmed the presence of 100 bp amplicon corresponding to the *EWSR1-ERG* fusion breakpoint in both the original patient DNA and the bioprinted sample (Fig. 3C). All three osteosarcoma tumours—zccs225, zccs43, and zccs265—

A) i)

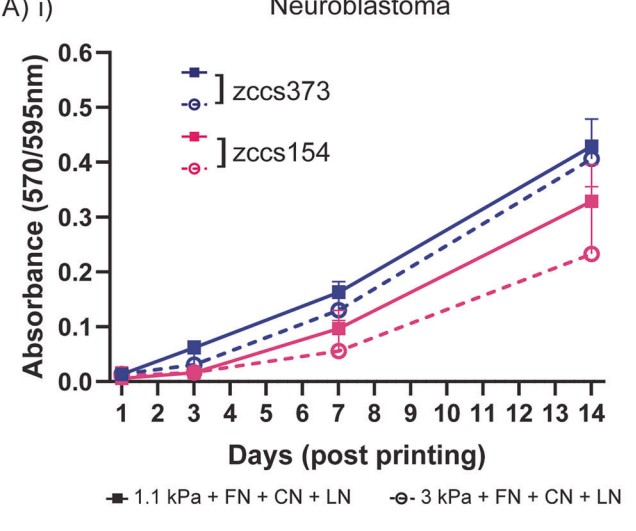

ii)

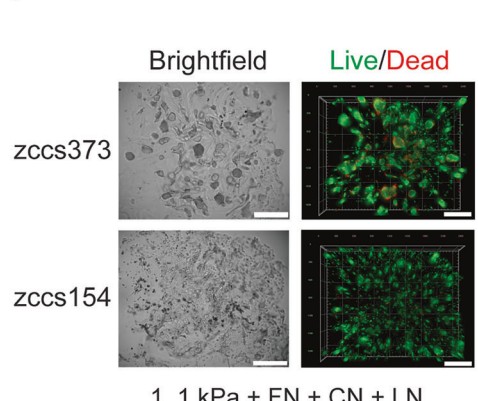

B) i)

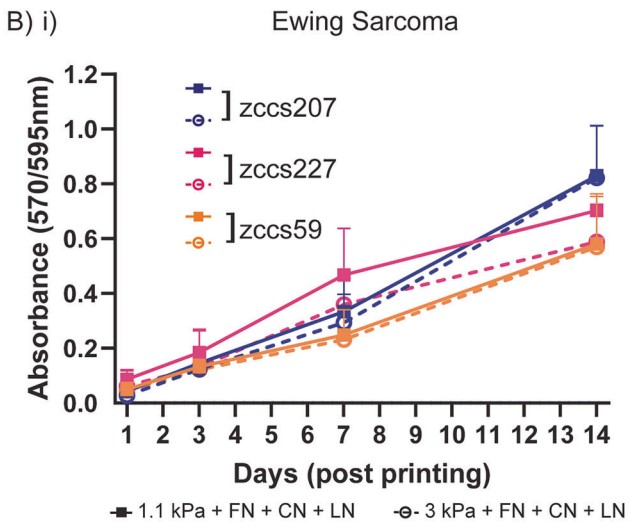

ii)

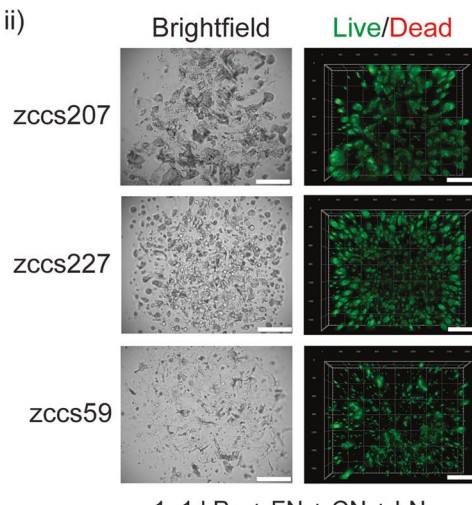

C) i)

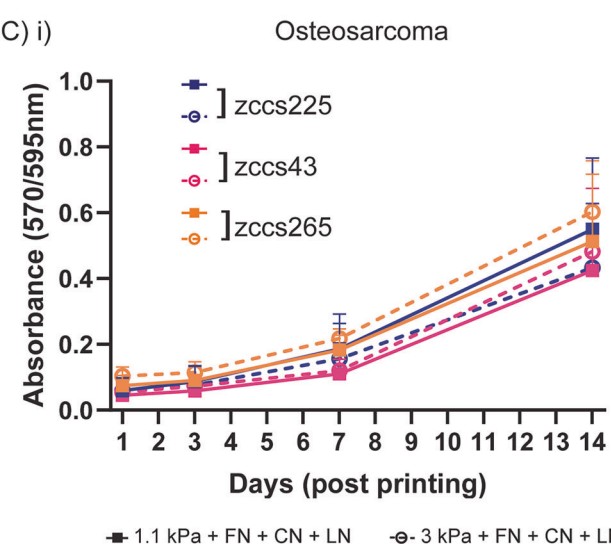

ii)

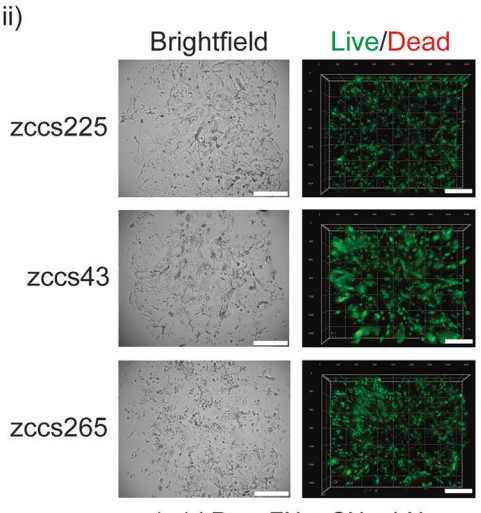

**Figure 2. Tunable ECM-mimic hydrogels support paediatric tumour growth.**

(A–C) (i) Cell proliferation of 3D bioprinted PDX tumouroids grouped by disease type: neuroblastoma (zccs373, zccs154), Ewing sarcoma (zccs207, zccs227, zccs59) and osteosarcoma (zccs225, zccs43, zccs265). Each sample was bioprinted in either 1.1 kPa or 3 kPa PEG hydrogels, both containing FN (fibronectin), CN (collagen 1) and LN (laminin) peptides. Proliferation rate was measured at day 1, 3, 7 and 14 post-printing. Data are presented as mean ± SD, n.s. $P = 0.9974$ (zccs373), $P = 0.9907$ (zccs154), $P = 0.9999$ (zccs207), $P = 0.9999$ (zccs59), $P = 0.9999$ (zccs225), $P = 0.9999$ (zccs43), $P = 0.9996$ (zccs265) (one-way ANOVA with Tukey's multiple comparisons test for each disease group). All experiments were repeated three times, except for zccs227, which had limited sample availability and is $n = 2$, therefore statistics are not presented for this sample. (A–C) (ii) Viability of 3D bioprinted samples in 1.1 kPa gels containing FN, CN and LN peptides, grouped by cancer type. Cells were stained with calcein-AM (green; live)/ethidium homodimer 1 (red; dead) Live/Dead Assay. Z stack 3D images were taken at day 14 post-printing. Representative images shown for each patient sample in brightfield (left) and Live/Dead (right). Scale bars on all images are 500 μm. Source data are available online for this figure.

harboured structural variations in *TP53*. Targeted sequencing confirmed the preservation of the *TP53-NUDT1* fusion in the matching bioprinted tumouroid for zccs225 (Fig. 3B). The original zccs43 patient sample contained an intragenic structural variant of *TP53* with a breakpoint in the noncoding region (exon 1), and the zccs265 sample included an intergenic *TP53-LSMD1* structural variant. The presence of these structural variants was not detected in corresponding PDX-derived tumouroids using targeted panel sequencing due to the technical limitations of the assay (Newman et al, 2021). Nevertheless, we subsequently confirmed the genetic match of the PDX-derived tumouroid to the original patient tumour using amplicon analysis. Bioprinted tumouroids zccs43 and zccs265 were characterised using both short tandem repeat (STR) and single-nucleotide polymorphism (SNP) profiling. STR profiles of bioprinted samples were 100% identity match to the original patient tumour. Additionally, SNP profiling confirmed high percentages of tumour cell content in the bioprinted tumouroids with overall ploidies and copy number profiles matching the original patient tumour (Fig. 3Di,ii). For zccs43, the tumour cell content was 92% with an overall ploidy of 1.72, compared to 86% and an overall ploidy of 1.64 in the original patient tumour (Fig. 3Di). Similarly, zccs265 exhibited a tumour cell content of 83% with an overall ploidy of 3.55, compared to 61% and an overall ploidy of 3.93 in the original patient tumour (Fig. 3Dii). Moreover, in osteosarcoma bioprinted sample zccs265, targeted sequencing confirmed the presence of *RB1* (p.Glu170ValfsTer6) nonsense variant and *BCL2* amplification (Fig. 3B). Taken together, our data indicate that 3D bioprinted PDX tumouroids retained critical pathogenic DNA variants and key molecular characteristics, confirming their close resemblance to the original patient tumours.

To further investigate the effect of bioprinting and culture on the molecular characteristics of the 3D tumouroids, we performed a comparative RNA-seq analysis on patient-derived cells from three representative PDX samples, both pre- and post-3D bioprinting (Fig. EV3). Bioprinted tumouroids exhibited a strong positive correlation in gene expression profiles with patient-derived xenograft (PDX) cells across neuroblastoma (zccs373, $r = 0.96$, $P < 0.001$), Ewing sarcoma (zccs207, $r = 0.99$, $P < 0.001$), and osteosarcoma (zccs225, $r = 0.98$, $P < 0.001$) samples (Fig. EV3Ai–Ci), confirming high molecular fidelity post bioprinting. We further analysed PDX and bioprinted samples for expression levels of disease-specific markers used in prognosis and diagnosis by the ZERO Personalised Medicine Team (Lau et al, 2024). In neuroblastoma (zccs373), LIN28B and PHOX2A showed comparable expression levels (Fig. EV3Aii). Similarly, BCL11B and GLG1 were assessed in Ewing sarcoma (zccs207), and CCNE1 and SPP1 in osteosarcoma (zccs225) (Fig. EV3Bii,Cii). These findings

indicate that key disease-specific molecular characteristics are preserved following 3D bioprinting. We next performed principal component analysis (PCA), which demonstrated that PDX samples and their corresponding bioprinted samples clustered closely, indicating a high degree of similarity in global gene expression profiles (Fig. EV3D). Furthermore, molecular aberrations previously identified by targeted panel sequencing (TSO500) were confirmed by RNAseq (Table EV5). Our findings confirm that bioprinted tumouroids faithfully replicate the molecular expression profiles of xenograft tumours, reinforcing our model's ability to retain original molecular characteristics after bioprinting.

## 3D bioprinted tumouroids retain primary phenotypic and tumourigenic characteristics

Next, the morphological features of the 3D bioprinted PDX tumouroids were characterised in comparison with the features of the patient-derived xenografts (PDXs) and the original patient samples. PDX cells were bioprinted and cultured for 7–14 days, prior to preparation and cryosectioning. Samples were stained with both routine H&E and tumour-specific immunohistochemical staining. The bioprinted tumouroids recapitulated the morphologic features of the original patient sample and PDX from which they were derived (Fig. 4A–C). Each of the bioprinted tumouroids contained "small round blue cells" in H&E staining, consisting of tumour cells with round to oval hyperchromatic nuclei and minimal cytoplasm (Fig. 4A–C, left panel). The proliferative capacity of the bioprinted cells was demonstrated by a high proportion of cells expressing Ki-67 (Fig. 4A–C, middle panel). The tumour type was confirmed by diagnostic immunohistochemical staining. The neuroblastoma tumouroids demonstrated diffuse strong nuclear PHOX2B (Hata et al, 2015; Trochet et al, 2004) staining (Fig. 4A, right panel), Ewing sarcoma tumouroids showed circumferential membrane staining for CD99 (Ambros et al, 1991; Zollner et al, 2021) (Fig. 4B, right panel), and osteosarcoma tumouroids with nuclear SATB2 (Conner and Hornick, 2013; Milton et al, 2022) staining (Fig. 4C, right panel). Quantification of Ki67 expression showed comparable levels between PDX tissues and 3D bioprinted tumouroids, indicating similar levels of proliferative activity in the cells post bioprinting (Fig. EV4A–C). These results confirm that the morphologic and diagnostic immunohistochemical characteristics of both original patient samples and PDXs are preserved in 3D bioprinted tumouroids.

Lastly, to ensure phenotypic features of the original cells were maintained, we examined tumourigenic capacity of PDX cells after 3D bioprinting and expansion. Two tumour types were examined for tumourigenic potential post bioprinting: one neuroblastoma

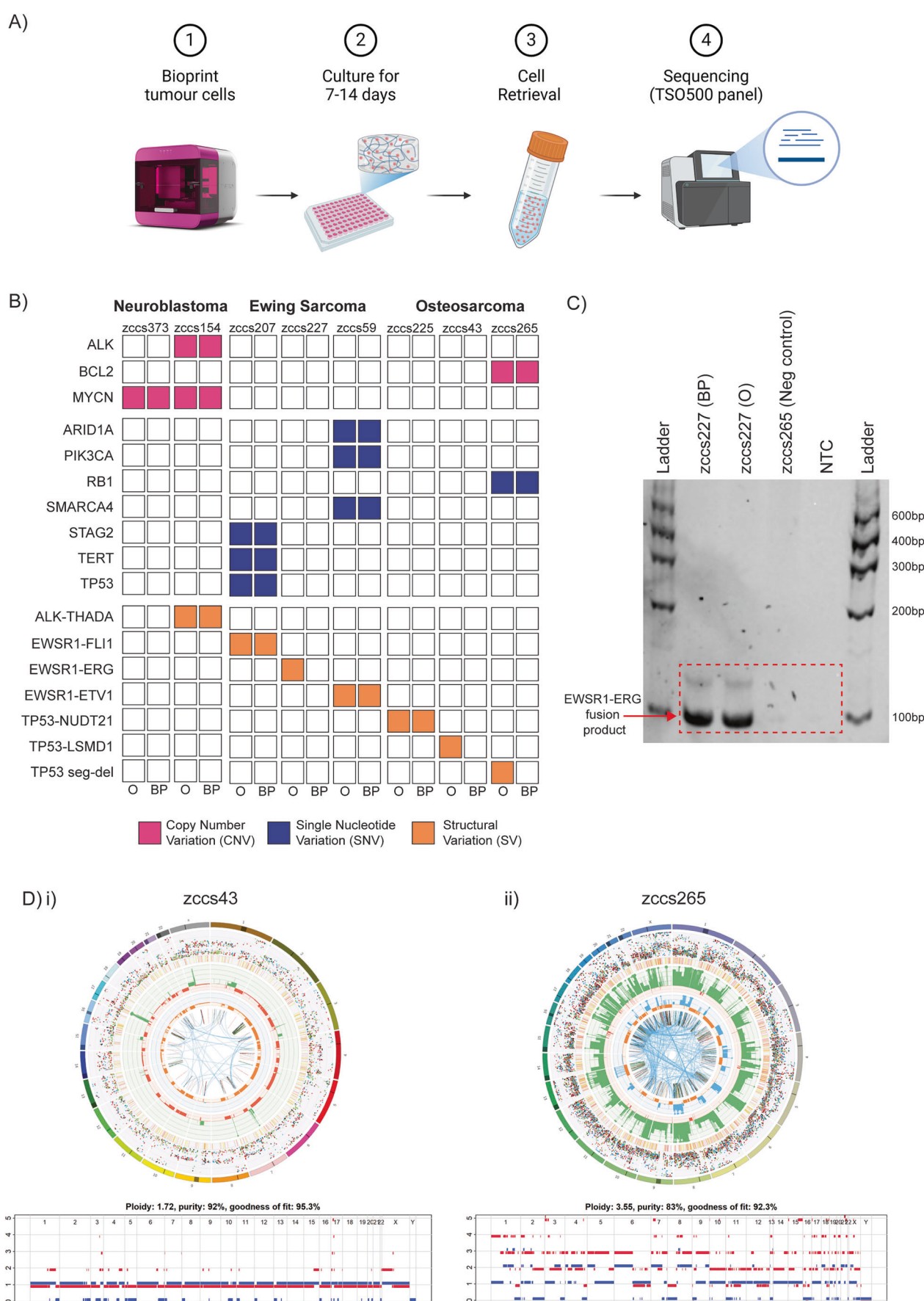

**Figure 3. 3D bioprinted tumouroids retain primary molecular characteristics.**

(A) Schematic showing workflow of obtaining bioprinted PDX tumouroid samples for sequencing. Samples were bioprinted in 1.1 kPa + FN + CN + LN hydrogels and cultured for 7–14 days, then cells were retrieved and purified for sequencing against the TruSight Oncology 500 panel. (B) Reportable pathogenic events for each patient-derived sample in either the Original patient (ZERO) (O) sample or the Bioprinted PDX (BP) sample, colour coded by mutation type. (C) PCR analysis of the DNA from bioprinted PDX sample zccs227 (BP) and the original patient zccs227 (O) confirms the presence of *EWSR1-ERG* structural variant. Zccs265 (osteosarcoma sample) used as negative control; NTC no template control. (D) (i–ii) Circos plots showing genome-wide profile of the original zccs43 and zccs265 osteosarcoma patient samples identified through whole genome sequencing (WGS) in comparison with ASCAT copy number profiles for bioprinted PDX samples generated from Single Nucleotide Profiling (SNP). Source data are available online for this figure.

(zccs373) and one Ewing sarcoma (zccs207). PDX cells dissociated from the bioprinted tumouroids were subcutaneously engrafted into immunocompromised NSG mice (*n* = 4 per group), while non-bioprinted in vivo expanded cells were used as a control (Fig. 5A). Both neuroblastoma (zccs373) and Ewing sarcoma (zccs207) tumour cell growth in vivo was comparable before and after bioprinting (Fig. 5Bi,Ci). Moreover, tumour morphology, Ki-67 proliferative index (Fig. EV4D,E) and tumour immunohistochemical staining for PHOX2B (neuroblastoma) and CD99 (Ewing sarcoma) remained unchanged between tumours generated from non-bioprinted and bioprinted cells (Fig. 5Bii,Cii). Taken together, these analyses confirm that 3D bioprinting conditions for high-throughput generation of tumouroids from neuroblastoma and sarcoma PDX cells do not alter the in vivo tumourigenic capacity of the cells.

## 3D bioprinted tumouroids are compatible with a high-throughput preclinical drug screening platform and reveal patient-specific drug vulnerabilities

Having confirmed our ability to grow and expand patient-derived PDX cells in an ECM-mimic environment while preserving their key molecular and phenotypic characteristics ex vivo, we proceeded to assess the compatibility and feasibility of 3D bioprinted tumouroids with the HTP ex vivo drug screening platform established by ZERO (Mayoh et al, 2023). We designed a custom-made drug library of 48 compounds selected from US Federal Drug Administration (FDA)-approved standard of care drugs for neuroblastoma and sarcoma patients, therapeutic agents previously identified by ZERO to be effective in specific high-risk paediatric samples (Lau et al, 2022) and some previously unconsidered targets of potential interest (Table EV6).

To initially assess the applicability of our 3D bioprinting approach for preclinical drug testing, we performed a side-by-side comparison of HTP drug screening using patient-derived xenograft cells, applying either the standard ZERO protocol (Mayoh et al, 2023) (2D or 3D spheroids in vitro culture) or the 3D bioprinting approach. Our findings demonstrate a high concordance in drug sensitivity profiles between the 3D bioprinted and ZERO preclinical testing protocols (Fig. EV5A–C). Both approaches reliably identified drug responses, with a strong correlation observed between the two approaches—for example, $r = 0.96$ for neuroblastoma zccs373, $r = 0.97$ for Ewing sarcoma zccs207, and $r = 0.98$ for osteosarcoma zccs225 samples (all $P < 0.001$) (Fig. EV5Ai–Ci). The key differentiator is that the platform enables standardised, scalable, and reproducible bioprinting of tumouroids in minutes, which is not feasible with manually seeded organoids. Heatmap clustering further confirmed similar sensitivity patterns across drug

classes, including both chemotherapeutics and targeted agents (Fig. EV5Aii–Cii). However, a consistent difference across three HTP screens was that 3D bioprinted cultures exhibited reduced sensitivity, with higher $IC_{50}$ values compared to ZERO (Table EV7). This suggests that lower drug concentrations were required to achieve 50% cell killing in the ZERO protocol compared to 3D bioprinted samples.

Having confirmed the feasibility of our bioprinting and HTP drug screening platform, we successfully screened six bioprinted patient-derived xenograft samples (Table EV8). Patient-derived xenograft cells were bioprinted, cultured between 7 and 14 days (depending on the patient-specific variability in cell growth) and subjected to an established ZERO pipeline for ex vivo drug screening using our customised library (Fig. 6A). To examine overall agent class susceptibility in our samples, we ranked the drugs by lowest median area under the curve (AUC) and log2 $[IC_{50}]$ values. From the analysis, we found that the most potent compounds for the six high-risk cancer patients were dactinomycin (RNA synthesis inhibitor); panobinostat (pan-HDAC inhibitor); dinaciclib (CDK1/2/5/9 inhibitor); SN-38 and GENZ-644282 (DNA topoisomerase I inhibitors); carfilzomib (proteasome inhibitor) and ceritinib (ALK inhibitor) (Figs. 6B and EV5D). This is consistent with findings across multiple paediatric preclinical drug screen studies which identified general responses across all tumour types to chemotherapeutic agent dactinomycin (actinomycin D) (Acanda De La Rocha et al, 2024; Peterziel et al, 2022) and targeted agents panobinostat and dinaciclib (Mayoh et al, 2023), carfilzomib (Acanda De La Rocha et al, 2024) and ceritinib (Peterziel et al, 2022).

To investigate tumour-specific drug sensitivity, T-score values of AUC and log2 $[IC_{50}]$ were compared based on the most differential drug responses in the cohort. We identified tumour-specific drug sensitivity to one or more library compounds (T-score ≤1) in four out of six successfully screened samples. Unique differential sensitivities to targeted agents were detected in two neuroblastoma and two Ewing sarcoma samples. Our analysis revealed that neuroblastoma zccs373 was most differentially sensitive to aurora-A kinase (AURKA) inhibitor alisertib and B-cell leukaemia/lymphoma 2 (BCL-2) inhibitor venetoclax (Fig. 6D). Interestingly, this sample did not contain molecular indicators for BCL-2 sensitivity. However, this sample had MYCN amplification (Fig. 3B), which could explain the observed sensitivity to both alisertib and venetoclax. Alisertib alters the conformation of AURKA, thereby disrupting its association with MYCN and inducing its ubiquitin-dependent degradation (Richards et al, 2016). MYCN amplification primes neuroblastoma cells by upregulating pro-apoptotic BCL-2 family member NOXA, creating vulnerability for BCL-2 inhibition by venetoclax (Ham et al, 2016).

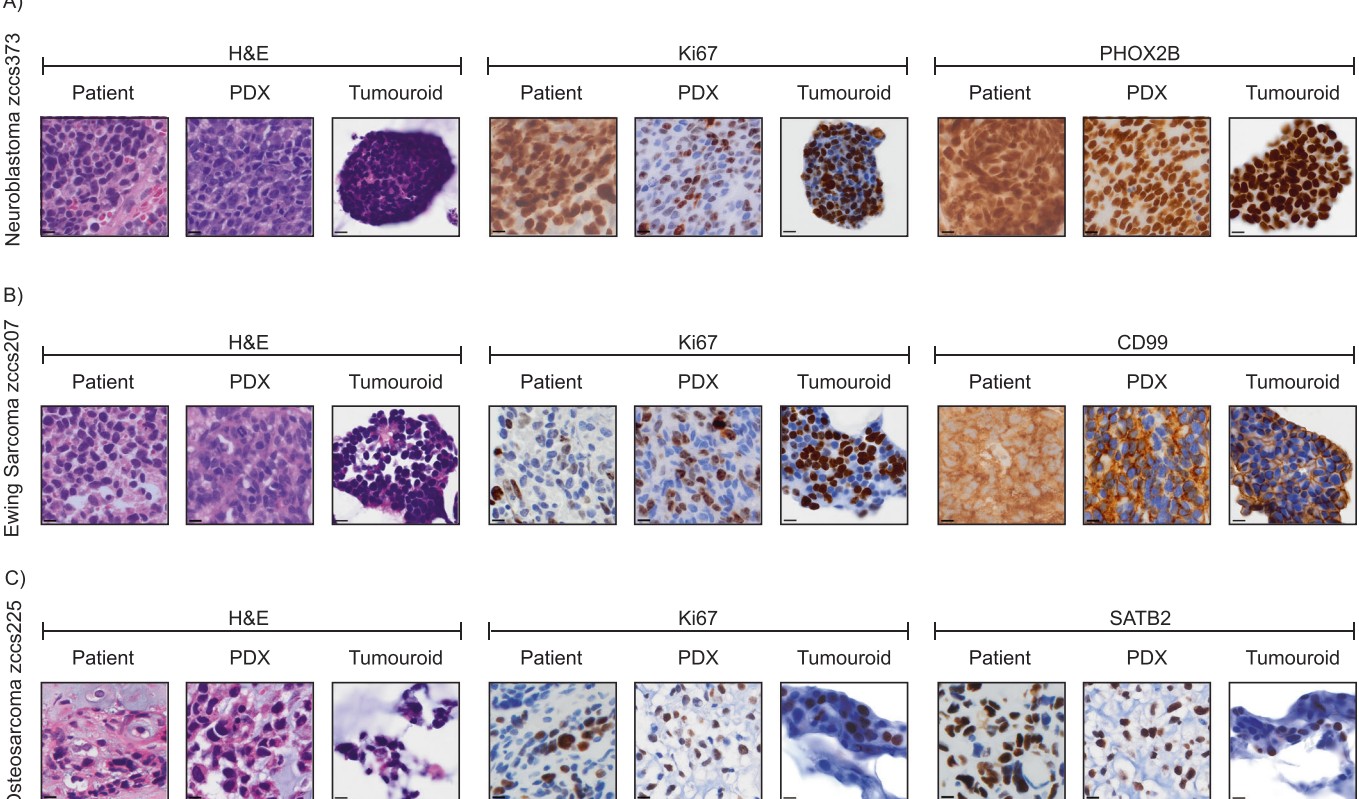

**Figure 4. 3D bioprinted tumouroids retain primary phenotypic characteristics.**

(A–C) Representative histology and immunohistochemistry images of the original patient tumours (patient), patient-derived xenografts (PDX) and bioprinted PDX tumouroids (tumouroid) from neuroblastoma (zccs373), Ewing sarcoma (zccs207) and osteosarcoma (zccs43). Samples stained with H&E (left), proliferation marker Ki-67 (middle) and tumour-specific markers (right) PHOX2B for neuroblastoma, CD99 for Ewing sarcoma and SATB2 for osteosarcoma. Scale bars are 10 μm for all images. Source data are available online for this figure.

Interestingly, another MYCN amplified neuroblastoma sample zccs154 (Fig. 3B) was less sensitive to both alisertib and venetoclax despite containing MYCN amplification (Table EV9). This sample was differentially sensitive to proteasome inhibitor carfilzomib (Fig. 6C). Even though molecular profiling identified ALK amplification (Fig. 3B) in this sample, it was not differentially sensitive to any of the ALK inhibitors included in our panel (alectinib, crizotinib, ceritinib) (Table EV6), confirming an established observation in personalised drug testing that the presence of molecular aberrations does not always translate into targeted drug sensitivities in vitro (Lau et al, 2022). The zccs207 Ewing sarcoma sample was differentially sensitive to several agents, including chemotherapeutics temozolomide, carboplatin and thiotepa. Targeted agents included several receptor tyrosine kinase, MAPK-ERK, PI3K pathway and cell cycle inhibitors (Fig. 6E). Ewing sarcoma sample zccs227 showed differential sensitivity to a number of chemotherapeutic agents including doxorubicin, irinotecan and vincristine as well as targeted agents affecting PI3K pathway, receptor tyrosine kinase (ALK) and cell cycle (AURKA) signalling pathways (Fig. 6F). Osteosarcoma samples zccs225 and zccs265 were generally sensitive to dactinomycin, dinaciclib (CDK1/2/5/9 inhibitor), carfilzomib (proteasome inhibitor) and ceritinib (ALK inhibitor) (Figs. 6B and EV5D). Compared to other samples in the cohort, no patient-specific drug

sensitivities in osteosarcoma samples were detected. This could be explained in part because many genetic drivers in paediatric osteosarcoma are transcription factors, epigenetic regulators, and fusion oncoproteins, which present significant challenges for targeting (Laetsch et al, 2021). We suggest that the application of a larger and more targeted compound library for osteosarcomas will be beneficial for expanding differential sensitivity readouts for this cancer type.

Next, we determined the feasibility of 3D bioprinting to expand original patient tumour samples for characterisation and drug screening. We accessed two cryopreserved high-risk Ewing sarcoma (EWS) patient samples—zccs486 and zccs1035 (Table EV1). Patient cells were bioprinted as described for Fig. 6. Initially, resazurin-based assays demonstrated low metabolic activity in both samples, followed by an increase in activity at day 20 for zccs486 (Fig. EV6A) and between days 7 and 14 for zccs1035 (Fig. EV6B). Live/dead imaging confirmed viability during culture for both samples (Fig. EV6C,D). Histological analysis of the zccs486 cells confirmed that the cells expanded post-bioprinting were proliferating (Ki67) and expressed key EWS disease-specific markers, including CD99 and NKX2.2 (Fig. EV6E).

Due to limited sample availability, we conducted a single drug screen experiment on Ewing Sarcoma zccs1035 using three concentrations of alisertib, alongside negative (DMSO) and positive

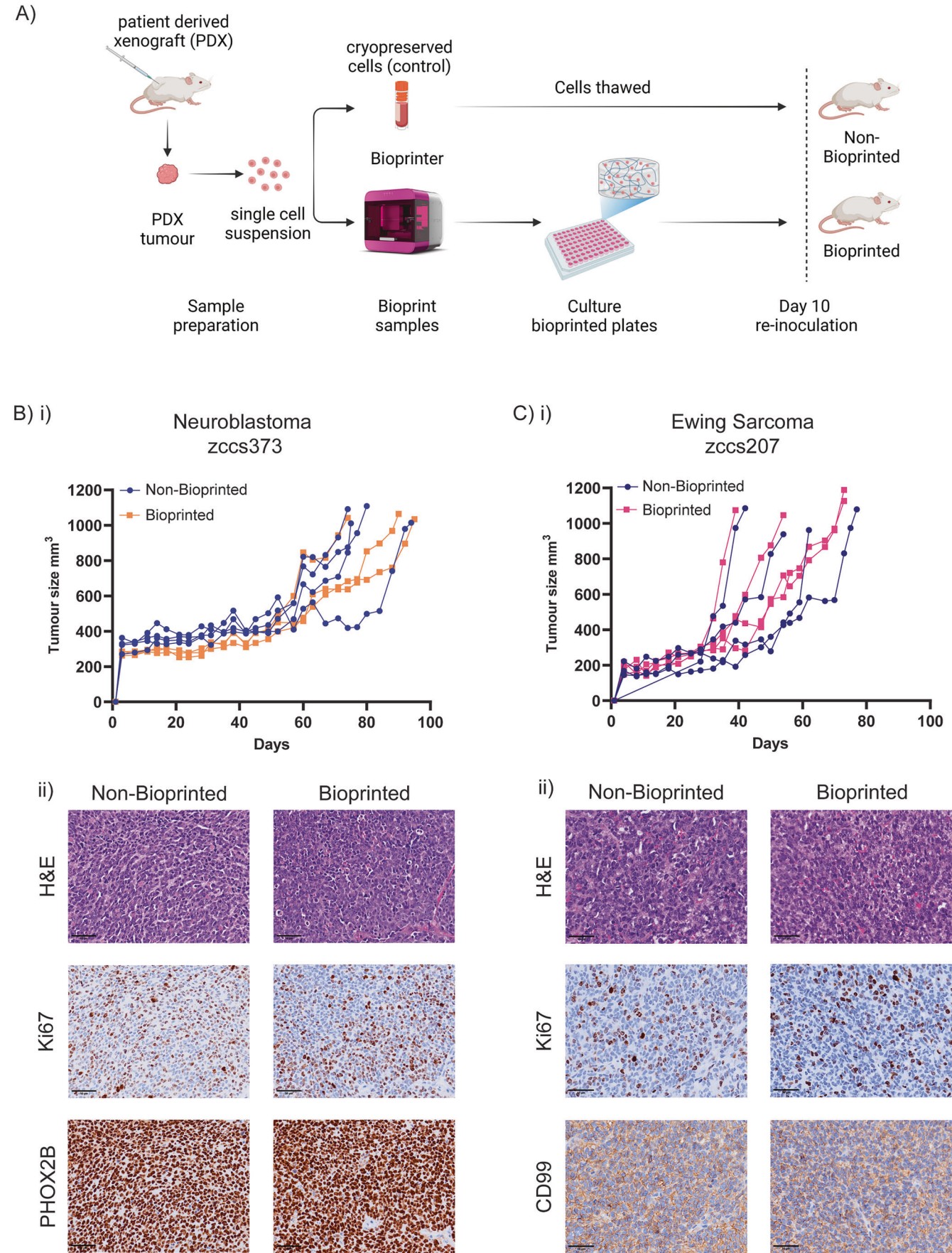

**Figure 5.  3D bioprinted tumouroids retain their primary tumourigenic characteristics.**

(A) Schematic workflow of in vivo tumourigenicity experiment. Briefly, one neuroblastoma and one Ewing sarcoma patient-derived xenograft tumour were dissociated to a single-cell suspension and bioprinted in 1.1 kPa + FN + CN + LN hydrogels for 10 days. At day 10, cells were retrieved and re-inoculated into NSG mice, with cryopreserved cells from the same tumour acting as control non-bioprinted samples. $n = 4$ mice per condition. (B, C) (i) Individual tumour growth curves for neuroblastoma and Ewing sarcoma tumours, respectively. One mouse was excluded from the zccs373 Bioprinted condition due to external injury. Data for each tumour type was analysed using a linear mixed-effects model with restricted maximum likelihood estimation (REML), n.s. $P = 0.3482$ (zccs373), $P = 0.6640$ (zccs207). (B, C) (ii) Representative histology and immunohistochemistry images from tumours derived from non-bioprinted (left) and bioprinted (right) PDX samples stained with H&E (top row), proliferative marker Ki-67 (middle row) and tumour-specific markers PHOX2B for neuroblastoma or CD99 for Ewing sarcoma (bottom row). Scale bars for all images are 50 µm. Source data are available online for this figure.

(100 µmol/L Benzethonium chloride) controls (Fig. EV6F). This data demonstrates the ability of the 3D bioprinted and expanded direct patient sample to respond to alisertib showing a trend towards a decrease in viability at 100 nM of drug compared to vehicle control. While the small sample size and variability limits the robustness of these findings, they provide preliminary evidence that direct bioprinting of patient samples for drug testing is feasible using our 3D bioprinting workflow.

Together, our analysis demonstrates the feasibility and compatibility of 3D bioprinted matrix-embedded tumouroids from patient-derived and patient cells with a HTP drug screening pipeline as well as the ability to identify differential patient-specific responses.

## Discussion

Precision medicine for childhood cancer that incorporates drug sensitivity profiling (DSP) can identify tailored therapies, avoiding the use of ineffective drugs and toxicity. Herein, we reveal the successful development of high-throughput (HTP) 3D bioprinted tumouroid models for high-risk paediatric tumours and their application in personalised DSP. Our models are uniquely engineered to be (i) reflective of the tumour-specific extracellular matrix environment; (ii) representative of the original patient tumours; (iii) expanded in a rapid manner; and (iv) compatible with high-throughput drug screening for timely therapeutic decision-making.

Our platform is innovative in its integration of defined biochemical and mechanical cues that mimic the native tumour microenvironment. We showed that addressing the tumour extracellular matrix requirements could inform culture conditions required for growth and expansion of non-epithelial tumour cells, which has previously proven to be challenging (Calandrini et al, 2020; Meister et al, 2022; Schott et al, 2024). This is the first study to demonstrate the incorporation of defined cell-adhesion peptides in the hydrogel systems for paediatric patient-derived cells in a high-throughput manner and apply this workflow in a preclinical drug testing pipeline. The hydrogels used in our study are highly tunable, well-defined and display reproducible stiffness and presentation of adhesive cues from batch-to-batch and are designed for use in a HTP 3D bioprinting platform (Du et al, 2023). Such properties could provide an advantage over the use of animal-derived ECM-like gels that have known limitations in replicating tissue-specific stiffness (Soofi et al, 2009) and variability associated with drug responses in cell culture-based experiments (Edmondson et al, 2016).

In paediatric preclinical cancer research, a significant challenge is acquiring enough cellular material for direct drug screening, mainly due to the limited amount of initial tumour material available (Mayoh et al, 2023; Peterziel et al, 2022). Propagating cells through patient-derived xenografts has emerged as a particularly vital strategy for cultivating tumour cells from paediatric patients (Kamili et al, 2020; Schott et al, 2024), however this approach can take months and is often unsuccessful, limiting its clinical value. An additional challenge is that embryonal tumours have a low mutational burden and are driven by a few key genetic events (e.g., MYCN amplification in neuroblastoma, RB1 loss in retinoblastoma). These events may not be sufficient to induce transformation in vitro unless the cells are in a very specific developmental state, making them difficult to grow and characterise (Custers et al, 2021). Adult carcinomas, on the other hand, accumulate mutations over time, which can make them more robust and easier to propagate in culture.

A major utility of our approach lies in its ability to generate tumouroids directly from patient-derived xenografts and cryopreserved patient samples within 7–20 days, bypassing the need for prolonged in vivo expansion. This significantly accelerates the timeline for preclinical testing and opens new possibilities for biobanking and personalised therapy development. Importantly, our bioprinted tumouroids retain the genomic, transcriptomic, and phenotypic fidelity of the original tumours, including key driver mutations, structural variants, and diagnostic markers. Successful expansion of fresh patient-derived xenograft or direct patient tumour cells in ECM-mimic hydrogels in a high-throughput fashion is an achievement rarely possible with conventional tissue culture. Indeed, we previously showed that HTP screening was only possible directly in 6% of paediatric fresh tumour samples and 26% were not considered for in vitro expansion as all material was required for molecular profiling (Lau et al, 2022). Of the remaining 68% that required expansion to gain a sufficient sample, only 22% underwent successful HTP screening and many failed expansion. Overall, HTP screening using either direct or expanded primary tumour cells was only feasible in 28% of fresh samples submitted. Our 3D bioprinting workflow demonstrated feasibility for cell expansion and potential biobanking in the ECM-mimicking environments.

The compatibility of our models with a high-throughput drug screening pipeline is a critical advancement. We demonstrated that the necessary cell numbers can be attained using our hydrogel systems, and high-throughput drug screening can be conducted directly in situ. Importantly, we were able to identify differential drug sensitivities in four out of six samples, all of which included one or more specific targeted agents.

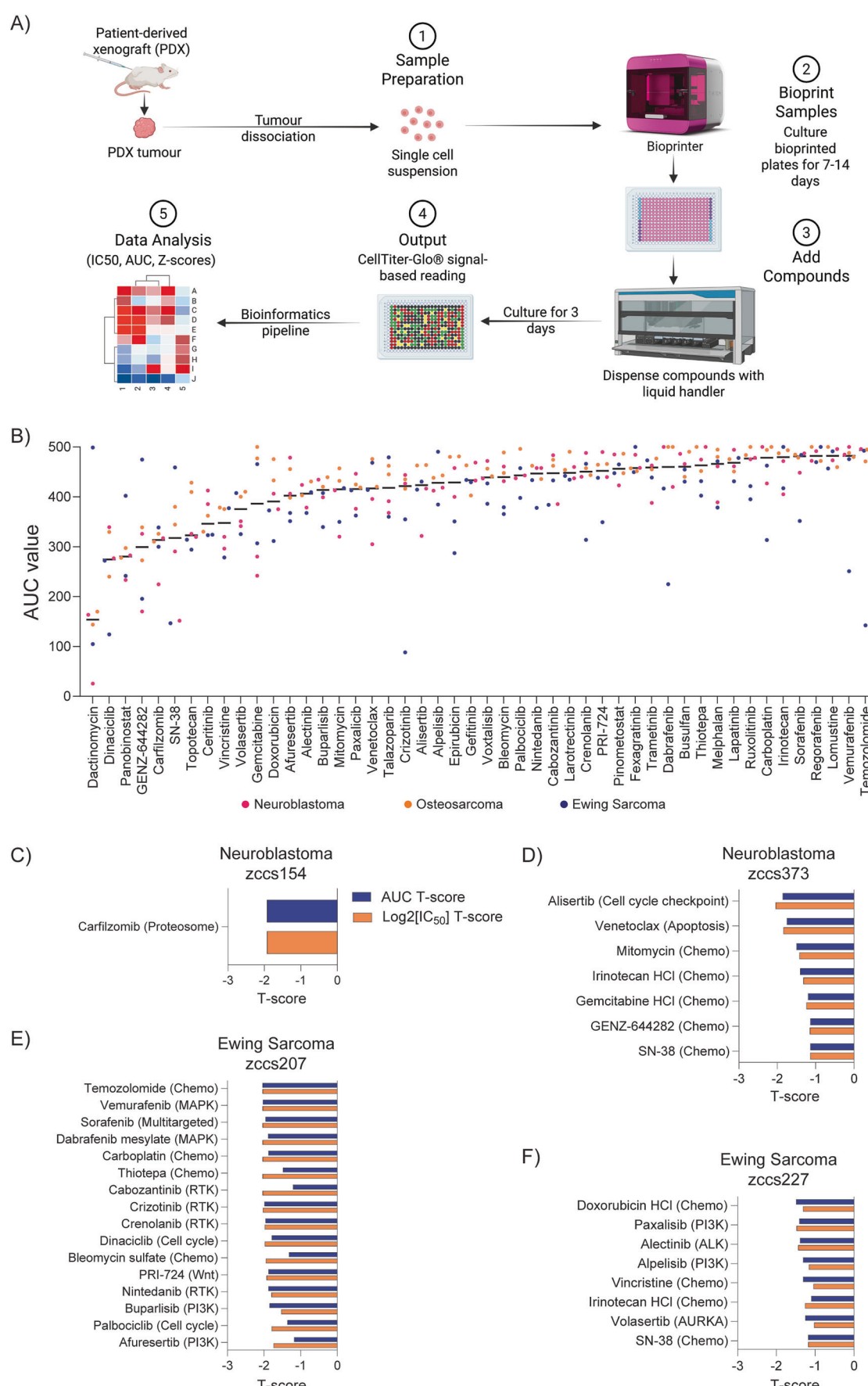

◀ **Figure 6. 3D bioprinted tumouroids are compatible with a high-throughput preclinical drug screening platform and reveal patient-specific drug vulnerabilities.**

(A) Schematic showing HTP drug screening pipeline. (B) AUC (area under the curve) distribution across all samples for the custom drug library of 48 compounds. Drugs are ordered from most (left) to least (right) effective based on the lowest median AUC. Colours indicate tumour type (pink—neuroblastoma, orange—osteosarcoma, blue —Ewing sarcoma). (C–F) Bar graphs representing tumour-specific drug sensitivity to drugs based on the combined AUC (blue) and log2 [IC$_{50}$] (orange) T-scores for each drug. T-score $\leq -1$ for both values is considered a differential response. Source data are available online for this figure.

Like the recently published HTP drug screening analyses of multiple paediatric patient cohorts, our study found that not all molecular vulnerabilities of patient-derived cells matched the specific drug susceptibilities identified in the ex vivo screens (Acanda De La Rocha et al, 2024; Dolman and Ekert, 2024; Peterziel et al, 2022). This suggests that our HTP drug screening in the 3D bioprinted tumouroid models closely resembles preclinical drug testing outcomes and is in line with clinical observations that targeting a genetic alteration with a targeted drug does not necessarily result in an objective patient response (Church et al, 2022; Lau et al, 2024). Our results confirm the importance of ex vivo HTP drug screening in support of clinical treatment decision-making for paediatric patients and suggest that 3D bioprinted HTP drug screens could identify previously unrecognised chemotherapeutic and targeted drug vulnerabilities independent of tumour molecular profile. However, clinical correlation of drug responses from bioprinted patient-derived xenograft tumouroids to patient outcomes has not been systematically investigated in this study, unlike non-bioprinted patient-derived organoids, where such correlations exist (Al Shihabi et al, 2024; Peterziel et al, 2022).

Compared to conventional organoid or spheroid models, our bioprinting workflow offers several advantages: automated and scalable production, defined ECM composition, and standardised culture conditions. These features enhance reproducibility and reduce variability, addressing key challenges in translational cancer research. While limitations remain, such as reliance on proprietary bioinks and the need for broader compound libraries, our platform sets the stage for future expansion across cancer types and integration into clinical trials. Future studies and clinical trials will investigate the relationship between patient outcomes and preclinical drug responses in 3D bioprinted patient models.

Our research demonstrates the application of the 3D bioprinting technology in creating patient-derived cancer models for downstream preclinical testing. Current advanced methodologies for preclinical 3D culture testing include scaffold-free spheroids (Acanda De La Rocha et al, 2024; Mayoh et al, 2023; Peterziel et al, 2022) and Matrigel or basement membrane extract (BME) cultures (Al Shihabi et al, 2024; Ding et al, 2022). These systems replicate key tumour features yet have critical limitations: (i) manually labour-intensive; (ii) either lack or have undefined ECM composition; (iii) suffer limited reproducibility (Aisenbrey and Murphy, 2020). Our HTP 3D bioprinting approach provides a robust, standardised, and high-throughput method for patient-derived tumour modelling. It allows scalable production of mini-tumour models in minutes in standardised ECM-mimic hydrogels with defined mechanical and biochemical properties. Our study is timely, as a recent review on tumouroids highlighted the need to optimise tumouroid protocols and analytical readouts to enhance the efficiency of tumouroid production and reduce assay turnaround times, thereby improving the speed and effectiveness of drug profiling (Kim et al, 2025).

In conclusion, our high-throughput 3D bioprinting approach represents a major advance in paediatric cancer modelling and drug testing. By enabling rapid, reproducible, and physiologically relevant tumouroid generation, we provide a powerful tool for precision oncology. This technology has broad applicability across developmental biology, cancer research, and therapeutic discovery, and holds promise for improving outcomes in children with high-risk cancers.

# Methods

## Reagents and tools table

| Reagent/resource | Reference or source | Identifier or catalogue number |
|---|---|---|
| **Experimental models** | | |
| Patient-derived xenografts (PDX) and cryopreserved direct patient samples | ZERO Childhood Cancer Precision Medicine Program | https://www.zerochildhood cancer.org.au/clinicians-researchers/for-researchers/access-to-zero-samples |
| Mouse: NOD.Cg-Prkdcscid Il2rgtm1Wjl/SzJAusb (NSG) | Australian BioResources | N/A |
| **Antibodies** | | |
| CD99 | BioCare Medical | #CME392A |
| Ki67 (SP6) | Thermo Fisher Scientific | #MA5-14520 |
| PHOX2B | Abcam | #ab183741 |
| SATB2 | Cell Marque | #CM384R15 |
| NKX2.2 | Abcam | #ab187375 |
| **Oligonucleotides and other sequence-based reagents** | | |
| Primers for EWSR-ERG fusion Forward: GACTGCTAGCCCTGCTG Reverse: GTAAAGGAGCTGTGTTGCTTAT | This paper | N/A |
| **Chemicals, enzymes and other reagents** | | |
| RPMI 1640 Medium, HEPES | Gibco | #22400089 |
| Growth Factor Reduced Matrigel | Corning | #354230 |
| Ammonium Chloride | Chem Supply (Westlab) | #AA049-500G |
| Potassium bicarbonate | Sigma-Aldrich | #237205 |
| EDTA disodium salt dihydrate | Chem Supply (Westlab) | #EA023-100G |
| Hydrochloric Acid | RCI Labscan | #AR1104 |
| IMDM | Gibco | #12440053 |
| MEM α, no nucleosides | Gibco | #12561056 |
| FBS | Gibco | #10100147 |
| Insulin-transferrin-selenium 100x | Gibco | #41400045 |
| ROCK inhibitor (Y-27632 2HCl) | Selleck Chemicals | #S1049 |
| Penicillin-streptomycin-glutamine (100X) | Gibco | #10378016 |
| Amphotericin B | Gibco | #15290018 |
| Hydrogel reagents | Inventia Life Sciences | #Px02.31P; #Px02.09P; #Px02.28P; #Px03.31P; #Px03.09P; #Px03.28P |
| Resazurin, sodium salt | Sigma-Aldrich | #199303 |
| Methylene blue | Sigma-Aldrich | #M9140 |

| Reagent/resource | Reference or source | Identifier or catalogue number |
|---|---|---|
| Potassium ferricyanide (III) | Sigma-Aldrich | #702587 |
| Potassium hexacyanoferrate (II) trihydrate | Sigma-Aldrich | #P9387 |
| Bovine serum albumin | Sigma-Aldrich | #A7906 |
| Cell Retrieval Solution | Inventia Life Sciences | #F235 |
| 0.3 M sodium acetate | Sigma-Aldrich | #241245 |
| 100% ethanol | Sigma-Aldrich | #459844 |
| 16% paraformaldehyde | ProSciTech | #C004 |
| O.C.T. compound | Tissue-Tek | #IA018 |
| Trypan blue solution, 0.4% | Gibco | #15250061 |
| HemosIL® Normal Control plasma | Instrumentation Laboratory | #0020003110 |
| Thromborel™ S | Siemens Healthiners | #10446445 |
| AmpliTaq Gold™ DNA polymerase | Applied Biosystems | #4311820 |
| ExoSAP-IT™ Express PCR Product Cleanup | Applied Biosystems | #75001.1.ML |
| **Software** | | |
| ActivityBase (v8.3.0.175) | IDBS | https://www.idbs.com/activitybase/ |
| Arivis Vision4D | Zeiss | https://www.zeiss.com/microscopy/en/products/software/arivis-pro.html |
| BBTools/BBMap v38.79 | Bushnell (2014) | https://sourceforge.net/projects/bbmap/ |
| ComplexHeatmap | Gu (2022) | N/A |
| CNVkit (v0.98) | Talevich et al (2016) | N/A |
| DNAnexus | | https://www.dnanexus.com/ |
| fastcluster | Müllner (2013) | N/A |
| Graph Prism (v.10.3.0) | GraphPad Software | https://www.graphpad.com/ |
| ImageJ | Schindelin et al (2012) | https://imagej.net/ij/ |
| Image Lab (v5.2.1) | Bio-Rad | https://www.bio-rad.com/en-au/product/image-lab-software |
| MatrisomeAnalyzeR | Petrov et al (2023) | https://sites.google.com/uic.edu/matrisome/tools/matrisome-analyzer |
| OncoKB | Chakravarty et al (2017) | N/A |
| PureCN (v1.16) | Riester et al (2016) | N/A |
| QPath | Bankhead et al (2017) | https://qupath.github.io/ |
| R (v.4.2.3) | N/A | https://www.r-project.org/ |
| TSO500 local app v2.0 | Illumina | https://support.illumina.com/content/dam/illumina-support/documents/documentation/software_documentation/trusight/trusight-oncology-500/trusight-oncology-500-local-app-v2-user-guide-1000000095997-02.pdf |
| Vardict (v1.8.2) | Lai et al (2016) | N/A |
| VEP (v100) | McLaren et al (2016) | N/A |
| Vcfanno | Pedersen et al (2016) | N/A |
| **Other** | | |
| Tumour Dissociation kit, human | Miltenyi Biotec | #130-095-929 |
| Mouse Cell Depletion kit | Miltenyi Biotec | #130-104-694 |
| Live/Dead viability/cytotoxicity kit | Invitrogen | #L3224 |
| All/Prep DNA/RNA Micro Kit | QIAGEN | #80284 |
| CellTiter-Glo® 3D Cell Viability Assay | Promega | #G9683 |
| Illumina Stranded Total RNA Prep with Ribo-Zero Plus kit | Illumina | #20040525 |

| Reagent/resource | Reference or source | Identifier or catalogue number |
|---|---|---|
| Whole genome sequencing (WGS) and RNA sequencing (RNASeq) patient data | ZERO Childhood Cancer Precision Medicine Program Wong et al (2020) | https://www.zerochildhoodcancer.org.au/clinicians-researchers/for-researchers/access-zero-data |
| Extracellular matrix organization Gene Set | Harmonizome Rouillard et al (2016) | https://maayanlab.cloud/Harmonizome/gene_set/Extracellular+matrix+organization/Reactome+Pathways |
| TSO500 and RNAsequencing PDX and bioprinted tumoroids data | The European Genome-phenome Archive (EGA) | https://ega-archive.org/EGAS00001008220 |
| Live/Dead imaging files | BioImage Archive | https://www.ebi.ac.uk/biostudies/bioimages/studies/S-BIAD2130 |

## Experimental model and study participant details

### Patient-derived samples

Patient-derived xenograft (PDX) models, patient DNA (zccs227) and cryopreserved patient samples (zccs1035 and zccs486) were obtained from ZERO Childhood Cancer Precision Medicine Program (ZERO). All procedures were approved by the Human Research Ethics Committee at the University of New South Wales (HC190693). We selected eight patient-derived xenograft models and two direct patient samples with a range of patient and disease characteristics (Table EV1).

### Animal studies

All animal studies and procedures were approved by the University of New South Wales Animal Care and Ethics Committee (ACEC 22/58B, ACEC 23/79B, ACEC 22/34B).

## Method details

### Extracellular matrix gene expression analysis

RNA expression data from high-risk neuroblastoma ($n = 41$); rhabdomyosarcoma ($n = 40$); Ewing sarcoma ($n = 40$) and osteosarcoma ($n = 25$) patients were extracted from the ZERO dataset (accessed on 1st of March 2024). Low-count genes (transcript count per million below 10 in half of the samples) were filtered out. ECM genes selected based on Extracellular Matrix Organization Gene Set ($n = 265$) extracted using Harmonizome (Rouillard et al, 2016). Extracellular matrix components were annotated with R (v4.2.3) and the MatrisomeAnalyzeR package (Petrov et al, 2023). 265 genes were ranked according to their transcripts per million (TPM) values, and the top and bottom 30 genes were selected. Z-scores for the TPM values were calculated and a heatmap with hierarchical clustering was generated with the ComplexHeatmap (Gu, 2022) and fastcluster (Müllner, 2013) packages in R. Further analysis was conducted on genes annotated as the core matrisome set ($n = 79$) with the MatrisomeAnalyzeR package. A comparative gene expression analysis was performed for collagen 1 (COL1A1) and fibronectin (FN1). Welch's $t$ test was used for statistical evaluation for both genes of interest, comparing these genes against the remaining core matrisome genes in the dataset.

### Patient-derived xenograft propagation

Briefly, 5–6-week-old female NOD.Cg-Prkdcscid Il2rgtm1Wjl/SzJAusb (NSG) mice were purchased from Australian BioResources (Moss Vale, NSW, Australia). The animals were housed in a pathogen-free environment in research animal cages with a total

volume of 400 cm² (Tecniplast, Italy) with air filters in positive pressure ventiracks. Environmental enrichment consisted of bedding, enviro-dry material, and igloos. Irradiated rat and mouse breeder cubes and water were supplied ad libitum. For inoculation with cryopreserved cells from previously dissociated patient-derived tumour xenografts, cells were suspended in serum-free RPMI medium (Gibco, #22400089) and combined with growth factor-reduced Matrigel® (Corning, #354230) at a 1:1 volume ratio (1 million cells per mouse). Cells were subcutaneously implanted into the flanks of female NSG mice using a 27-gauge needle (Terumo, Tokyo, Japan). For tumour fragment implantation, mice were anaesthetised, and a 5-mm horizontal incision was made on the dorsal surface, 10 mm anterior to the base of the tail, to create a subcutaneous pocket. A 2–5-mm tumour fragment, pre-soaked in growth factor-reduced Matrigel®, was inserted into the pocket, and the incision was closed with a wound clip. Buprenorphine (0.1 mg/kg, intraperitoneal) (Provet, Castle Hill, Australia) was administered for analgesia. Tumour size was monitored twice weekly using Vernier callipers, calculated using the formula: Length × width × height/2. Mice were euthanised when the tumour size reached 1000 mm³ by $CO_2$ overdose followed by cervical dislocation. Tumour tissues were then collected for tumour dissociation, bioprinting and histology studies.

### Tumour dissociation

Tumours were dissociated using tumour dissociation kits according to the manufacturer's instructions (Tumour Dissociation kit, human; Miltenyi Biotec, #130-095-929). Briefly, tumours were minced in RPMI media (Gibco, #22400089) and transferred to gentleMACS tubes with enzymes A, H and R to be homogenised using gentleMACS Octo Dissociator. Cell suspension was strained through a 70-μm strainer and spun down at 1500 rpm for 5 min (acceleration/deceleration speed 6) before supernatant was removed. Red blood cells were lysed by resuspending the cell pellet in 1× ACK Lysing buffer (155 mM Ammonium Chloride, 10 mM Potassium Bicarbonate, 0.1 mM EDTA, pH adjusted 7.2–7.4 with HCl) for 5 min, then spun down and supernatant discarded. The final pellet was resuspended in appropriate culture media and strained through a 70-μm strainer to achieve a single-cell suspension for cell counting and downstream experiments. Cell suspension was stored at 4 °C while being used.

### Mouse cell depletion

Mouse cell depletion was conducted according to the manufacturer's instructions for Mouse Cell Depletion kit (Miltenyi Biotec, #130-104-694). Briefly, a single-cell tumour suspension was obtained by passing through a 70-μm strainer and pelleting cells. Per 2 million tumour cells, pellet was resuspended in 80 μL of 0.5% bovine serum albumin (BSA; Sigma-Aldrich, #A7906) in DPBS and mixed with 20 μL of Mouse Cell Depletion cocktail then incubated for 15 min at 4 °C. Volume was adjusted to 500 μL using 0.5% BSA in DPBS and ran on autoMACS Pro Separator to magnetically separate positive fraction (mouse cells and dead cells) and negative fraction (human tumour cells).

### Patient-derived cells

All bioprinting experiments were carried out using freshly isolated patient-derived cells obtained from tumour xenografts (see "Patient-derived xenograft propagation" section above) or from cryopreserved patient tumour tissues. Patient-derived neuroblastoma cells were

cultured in vitro in IMDM media (Gibco, #12440053) containing 20% foetal bovine serum (FBS) and 1% Insulin-Transferrin-Selenium 100x (ITS-G, Gibco, #41400045). Sarcoma cells were cultured in alpha MEM (Gibco, #12561056) or IMDM media with 10% FBS, 1% ITS-G and 0.1% ROCK inhibitor (Y-27632 2HCl, SelleckChem, #S1049). All cells in vitro were cultured with 1% penicillin/streptomycin (Gibco, #10378016) and 0.1% Amphotericin B (Gibco, #15290018) and cultured in 5% $O_2$ 37 °C/$CO_2$ incubator.

### Bioprinting

Bioinks and activators for multiple hydrogels were sourced from Inventia Life Science, Sydney, Australia (see Table for further details). Hydrogels contain ECM-mimicking peptides of fibronectin (FN; RGD peptide), collagen 1 (CN; GFOGER peptide) or laminin (LN; DYIGSR peptide). 3D bioprinted models were printed using the RASTRUM™ 3D bioprinter (Inventia Life Science) as described previously (Utama et al, 2021). The RASTRUM™ 3D bioprinter utilises two elements: bioink and activator. Cells are resuspended in activator solution and loaded into the bioprinter, along with bioink, in separate reservoirs within the printer cartridge. Both solutions are bioprinted and form an instant gel when combined in a microplate. Printing protocols and hydrogel design were generated using RASTRUM™ Cloud (Inventia Life Science). In this study we utilised four bioprinting models: Imaging Model ($25 \times 10^6$ cells/mL; ~8000 cells printed per well), Immunohistochemistry Model ($25 \times 10^6$ cells/mL, ~16,000 cells printed per well), Large Plug Model ($12.5 \times 10^6$ cells/mL; ~36,000 cells printed per well); HTP Model ($2.04–4.08 \times 10^6$ cells/mL; ~2000–4000 cells printed per well) These seeding densities were used for all patient-derived samples except for zccs227, where the concentration of cells per ml was doubled for all bioprinting models. For cell viability and in situ imaging experiments, Imaging Model was used, and samples were bioprinted in standard flat-bottom 96-well plates (Greiner, #655180). The Immunohistochemistry Model was used for immunohistochemistry experiments and bioprinted in standard 24-well plates (Corning, #3524). For sequencing and tumourigenicity experiments, Large Plug Model was used, and samples were bioprinted in standard 96-well plates (Greiner, #655180). The HTP Model was used for drug screening, and cells were bioprinted in white clear bottom 384-well plates (Greiner, #781098).

| Hydrogel* | Stiffness | Mimicking component | Peptide/s | Catalogue code |
|---|---|---|---|---|
| 1.1 kPa + FN | 1.1 kPa | Fibronectin (FN) | RGD | Px02.31P |
| 1.1 kPa + FN + CN | 1.1 kPa | Fibronectin (FN) Collagen 1 (CN) | RGD GFOGER | Px02.09P |
| 1.1 kPa + FN + CN + LN | 1.1 kPa | Fibronectin (FN) Collagen 1 (CN) Laminin (LN) | RGD GFOGER DYIGSR | Px02.28P |
| 3 kPa + FN | 3 kPa | Fibronectin (FN) | RGD | Px03.31P |
| 3 kPa + FN + CN | 3 kPa | Fibronectin (FN) Collagen 1 (CN) | RGD GFOGER | Px03.09P |
| 3 kPa + FN + CN + LN | 3 kPa | Fibronectin (FN) Collagen 1 (CN) Laminin (LN) | RGD GFOGER DYIGSR | Px03.28P |

*Hydrogels were obtained from Inventia Life Science and contain ECM-mimicking peptides of fibronectin (FN; RGD peptide), collagen 1 (CN; GFOGER peptide) or laminin (LN; DYIGSR peptide).

### Cell viability and live/dead imaging

Tumour samples were bioprinted in each hydrogel condition using Imaging Models and cultured in 5% $O_2$ 37 °C/$CO_2$ incubator for up to 14 days. Media was replenished on days 3, 7 and 10. Cells were incubated with resazurin-based reagent (Sigma-Aldrich, #199303) at 10% media volume for 24 h on days 1, 3, 7 and 14. The assay was read at 570–595 nm using the Benchmark Plus plate reader (Bio-Rad), and absorbance values were normalised to Ethanol-treated control wells (blank negative control). For Live/Dead imaging, cells were stained with Live/Dead viability/cytotoxicity kit, for mammalian cells (Invitrogen, #L3224) according to the manufacturer's instructions and previously published (Jung et al, 2022). Briefly, cells were bioprinted and cultured for up to 14 days. At day 14, hydrogels were rinsed with DPBS and stained with 100 µL of live/dead staining solution (10 µM Ethidium Homodimer-1 and 5 µM Calcein AM in DPBS), then incubated at 37 °C for 30–45 min. Z-stack images were taken at ×5 magnification using green fluorescence for live cells and red fluorescence for dead cells on the CellDiscoverer 7 microscope (Zeiss). Images were visualised using Arivis Vision4D software (Zeiss).

### Cell extraction

Cells were collected using the RASTRUM Cell Retrieval Solution, purchased from Inventia Life Science (#F235), according to the manufacturer's instructions. Cells were bioprinted in Px02.28 P hydrogels and cultured until endpoint. Hydrogels were rinsed with DPBS and incubated with 50 µL Cell Retrieval Solution for 30 min at 37 °C/5% $CO_2$. Solutions were vigorously resuspended to dislodge cells from hydrogels, and supernatant was collected in a 50 mL centrifuge tube through a 70-µm strainer to remove any hydrogel fragments. Wells were rinsed with DPBS and contents combined with the collected cell suspension. Cells were pelleted at 1200 rpm for 5 min, then pellets were rinsed twice with cold DPBS.

## Sequencing and sample validation

### Targeted sequencing (TSO500)

Patient-derived cells were bioprinted and cultured for 7 or 14 days in relevant media, then cells were retrieved (see "Cell extraction" section above) and pellets were snap-frozen on dry ice and stored at −80 °C for downstream analysis. Cell pellets were sent to Central Adelaide Local Health Network Laboratories, where DNA was extracted and sequenced using the Illumina TruSight Oncology 500 (TSO500) panel. DNA fastqs were first aligned to human and mouse genome simultaneously by bbsplit from BBTools (v38.79), outputting reads best mapping to each genome to two interleaved fastqs respectively. Human genome interleaved fastqs were converted to paired read fastqs by reformat from BBTools (v38.79) and used as input for TSO500 app. For the TSO500 variant, calling was performed using the Illumina TSO500 local app (v2.0). For other panels, variant calling was performed using Vardict (v1.8.2). After variant calling, variant annotation was performed using VEP (v100) and annotated with COSMIC (v95), ClinVar (2020-05-13), gnomAD (r2.1.1) and CADD (v1.6) using vcfanno. Potential pathogenic variants were identified using these databases and confirmed in OncoKB. CNV estimation was performed using CNVkit (v0.98), and tumour purity and ploidy were estimated using PureCN (v1.16) to turn fold change into estimated copy number. The pipeline was run on DNAnexus and genomic interpretation finalised using Gentian, an in-house developed tool for assimilating and assessing targeted capture panel data.

### STR and SNP validation

For samples requiring further validation, DNA was extracted from bioprinted cell pellets using the AllPrep DNA/RNA Micro Kit (QIAGEN, #80284) and DNA concentrations measured using a NanoDrop spectrophotometer. DNA precipitation was performed to further remove buffer salts by incubating the sample DNA with 0.3 M Sodium Acetate (Sigma-Aldrich, #241245) and 2 volumes of 100% ethanol (Sigma-Aldrich, #459844) for 1 h at −20 °C. Samples were then rinsed in ice-cold 70% ethanol and resuspended in EB buffer (QIAGEN). DNA samples were submitted to the Garvan Institute of Medical Research for short tandem repeat (STR) profiling using the PowerPlex® 18D system, which analyses 18 markers. The results were compared with STR profiles from the original tumour sample. Profiles with more than 80% identity were considered a match. Additionally, DNA samples were submitted to the Victorian Clinical Genetics Services for single-nucleotide polymorphism (SNP) profiling using the Illumina Infinium Global Screening 531 Array-24 v2.0. The process involves two output metrics: Log R, indicating the overall signal intensity for both alleles, and B allele frequency (BAF), reflecting the allelic imbalance of single-nucleotide polymorphisms (SNPs). To generate genome-wide allele-specific copy number profiles, the R package ASCAT (v2.5.2) was utilised for data analysis. Validation of the results was conducted by comparing the derived copy number profiles and purity with those of the original patient tumour samples.

### PCR

Zccs227 patient-specific DNA sequence of EWSR-ERG breakpoint structural variant was obtained using WGS data. Custom oligonucleotide primers were designed using Integrated DNA Technologies PrimerQuest™ Tool (forward: GACTGC-TAGCCCTGCTG; reverse: GTAAAGGAGCTGTGTTGCTTAT). Standard PCR was performed using AmpliTaq Gold™ DNA Polymerase (Applied Biosystems, #4311820) at a setting of 40 cycles and annealing temperature of 60 °C on the Bio-Rad T100 Thermo Cycler. PCR products were visualised against an in-house molecular ladder on an in-house 12.5% polyacrylamide gel using the Gel Doc XR+ System and Image Lab Software v5.2.1 (Bio-Rad). PCR products were cleaned up with ExoSAP-IT™ Express (Applied Biosystems, #75001.1.ML).

### RNA sequencing

Single-cell suspensions of patient-derived samples were obtained from xenograft tumours (see "Tumour dissociation" section above). In total, $5 \times 10^6$ cells were centrifuged at 1200 rpm for 5 min, and pellets were rinsed twice with cold DPBS. Pellets of pre-bioprinted cells were snap-frozen on dry ice and stored at −80 °C. Remaining patient-derived cells were then bioprinted in Large Plug models and cultured for 7 days in relevant media. Cells were retrieved (see "Cell extraction" section above) and $5 \times 10^6$ cells were centrifuged at 1200 rpm for 5 min and pellets rinsed twice with cold DPBS before being snap-frozen on dry ice and stored at −80 °C for downstream analysis. Each patient sample was collected in triplicate (before and after bioprinting) from replicate xenograft tumours. Cell pellets

were sent to Australian Genome Research Facility (AGRF) where RNA was prepared using the Illumina Stranded Total RNA Prep with Ribo-Zero Plus kit (Illumina, #20040525), sequenced on the Illumina NovaSeqX as paired-end 150 base pair reads to a target depth of 100 million reads per sample. RNA fastqs were aligned to the human genome reference (hg38 and gencode v41) using STAR (v2.7.8a) in two-pass mode with –quantMode set to Transcripto-meSam, followed by alignments being sorted and duplicates marked with Picard Tools (v2.25.7). Fusions were identified using Arriba (v2.4.0). Genes were quantified using RSEM (v1.3.3) to obtain raw gene counts and transcripts per kilobase million (TPM). Differential expression, Pearson's correlation, principal component analysis (PCA) and gene expression pairwise analysis (using $t$ test statistic for significance) were all performed in R (v4.2.3). Differential expression analysis was performed using edgeR, PCA using ggfortify. Plots were generated in R using ggplot2.

### In vivo tumourigenicity

To determine if the 3D bioprinted and expanded patient-derived cells maintained their phenotypic characteristics to form tumours, we performed a tumourigenicity study as follows. Neuroblastoma zccs373 or Ewing Sarcoma zccs207 samples were propagated as described above and monitored until tumours reached endpoint at 1000 mm³. Tumours were collected, dissociated (see "Tumour dissociation" section above) and bioprinted in Large Plug hydrogels at 36,000 cells per well. Remaining tumour cells were cryopreserved in 90% FBS and 10% DMSO to be used as controls for tumourigenicity re-engraftment. Bioprinted samples were cultured for 10 days in relevant media before cells were extracted (see "Cell extraction" section above). Freshly extracted bioprinted cells and thawed, matched cryopreserved PDX cells were resuspended in 100% Growth Factor Reduced Matrigel® (Corning, #354230) and subcutaneously inoculated at 1 million cells per mouse for each condition into the flanks of female (5–7 weeks old) NSG mice (NOD.Cg-Prkdcscid Il2rgtm1Wjl/SzJArc) obtained from Austra-lian Resource Centre (Murdoch, Perth Western Australia). Once inoculated, tumour growth was monitored twice weekly using vernier callipers using the formula: $\frac{1}{2} \times L \times W^2$. Mice were euthanised when the tumour size reached 1000 mm³ by $CO_2$ overdose followed by cervical dislocation. Tumour tissues were then collected for histology and immunohistochemistry studies.

### Immunohistochemistry

Patient-derived cells were bioprinted in Px02.28 P hydrogels using the Immunohistochemistry Model and cultured for 14 days in relevant culture media. Bioprinted plates were fixed in 4% paraformaldehyde (PFA) in DPBS for 30 min at room temperature, then rinsed and stored in DPBS until use. For sample processing, hydrogels were first stained with 1:100 Trypan Blue (Gibco, #15250061) in DPBS for 15 min. Structures were then carefully scooped out of plates and transferred to cryomold cassettes filled with O.C.T. compound (Tissue-Tek, #IA018) before being snap-frozen on dry ice. Cryopreserved samples were processed by the Biological Specimen Preparation Laboratory in Katharina Gaus Light Microscopy Facility (KG-LMF) at the University of New South Wales into 5-μm sections. Patient-derived xenograft tumours were collected at endpoint (1000 mm³) (see "In vivo tumourigeni-city" section above), fixed in 10% formalin and paraffin-embedded. Samples were cut into 5-μm sections. H&E and antibody marker

staining were performed by the Histopathology department at the Garvan Institute of Medical Research using the Leica Bond system. For direct patient samples, cells were extracted (see "Cell extraction" section above) and spun at $300 \times g$ for 5 min, before all supernatant was removed. The cell pellet was resuspended in HemosIL® Normal Control plasma (Instrumentation Laboratory, #0020003110), then equal volumes of Thromborel™ S (Siemens Healthiners, #10446445) were added and allowed to clot at room temperature (10–15 min). The resulting cell plug was fixed in 10% neutral buffered formalin, paraffin-embedded and processed by Prince of Wales Hospital Anatomical Pathology Laboratory. Tumour-specific markers were visualised with CD99 (BioCare Medical, #CME392A; 1:50), PHOX2B (Abcam, #ab183741; 1:1000), NKX2.2 (Abcam, #ab187375; 1:100) and SATB2 (Cell Marque, #CM384R15; 1:100) antibodies. Ki-67 (SP6) (Thermo Fisher, #MA5-14520; 1:500) was used for cell proliferation.

### High-throughput drug screening

Patient-derived cells were bioprinted for drug screening (see "Bioprinting" section), and 50 μL of the relevant media was added to each well. Prior to bioprinting, sarcoma samples were subject to a mouse cell depletion protocol (see "Mouse cell depletion" section). All samples were printed in two identical plates (duplicates). Media was exchanged and replaced with 50 μL media every 3–4 days for the duration of the experiment. ZERO HTP drug screen experiments were set up according to methods previously described (Lau et al, 2022). Briefly, sarcoma cells were seeded into white ultra-low attachment 384-well plates (PrimeSurface, #MS-9384W) at 2000 cells/well and neuroblastoma cells were seeded into white flat-bottom 384-well plates (Greiner, #781080) at 2000 cells/well. All samples were seeded in two identical plates (duplicates) in a volume of 25 μL of relevant media. After 7 or 14 days of incubation for the bioprinted plates, or 3 days of incubation for the ZERO plates, a library of 48 compounds (Compounds Australia) was administered to duplicate HTP plates using the Hamilton STAR liquid handling robot in final concentrations ranging from 0.5 nmol/L to 5 μmol/L. Drug compounds were selected based on previously identified differential drug sensitivities in specific patient samples (Lau et al, 2022; Mayoh et al, 2023), standard of care chemotherapy agents against neuroblastoma and sarcoma tumours and included previously unconsidered targets of potential interest. Benzethonium chloride (100 μmol/L) and DMSO were adminis-tered as positive and negative controls, respectively. Seventy-two hours after drug administration, metabolic activity was measured using the CellTiter-Glo® 3D Cell Viability Assay (Promega, #G9683). For bioprinted plates, 50 μL of CellTiter-Glo® reagent was added to each well (1:1 to media volume) and shaken on plate shaker at 450 rpm for 2 h, protected from light. For ZERO plates, 25 μL of CellTiter-Glo® reagent was added to each well (1:1 to media volume) and shaken for 1 min. Neuroblastoma samples were incubated for 5 min, while sarcoma samples were incubated for 2 h. Both samples were protected from light. Following incubation, ZERO plates were shaken for 5 min on the EnVision® plate reader (PerkinElmer) before luminescence was read for 1 s per well. Luminescence was read immediately after incubation time for the bioprinted plates. Quality control cut-offs were defined based on two key metrics: (1) a Pearson correlation coefficient >0.7 between technical replicate plates, and (2) a robust Z'-factor >0.3 for both replicates.

A robust Z-prime was calculated based on the following formula:

$$Z'_{robust} = 1 - \frac{3 \times \left(MAD_{positive} + MAD_{negative}\right)}{\left|Median_{positive} - Median_{negative}\right|}$$

MAD is a Median Absolute Deviation. One sample (zccs43) failed post-screen quality control due to high variability in luminescence values between replicates, and one sample (zccs59) failed due to insufficient cellular outgrowth prior to drug screening.

Data analysis was performed as previously described (Mayoh et al, 2023). ActivityBase (IDBS) software (Version 8.3.0.175) was used to generate cell viability percentages from raw luminescence data using the following formula: ([readout value drug − average readout of positive controls]/[average readout of negative $^{controls}$ − average readout of positive controls]) × 100. Calculated viabilities were used to generate dose-response curves, calculate area under the curve (AUC) and half-maximal inhibitory concentration (IC$_{50}$) values.

### Direct patient samples

Cryopreserved direct Ewing sarcoma patient samples were obtained from ZERO Childhood Cancer Precision Medicine Program (ZERO). Zccs486 and zccs1035 samples were thawed and resuspended in relevant media, then incubated at 37 °C for 3 h to allow cells to recover. Patient-derived cells were bioprinted in HTP Model at 4000 cells/well in 384-well plates (see "Bioprinting" section), and 40 µL of the relevant media was added to each well. Media was exchanged and replaced with 40 µL media every 3–4 days for the duration of the experiment. Cells were incubated with resazurin-based reagent (Sigma-Aldrich) at 10% media volume for 24 h on days 1, 3, 7 and every 7 days until endpoint. The assay was read at 570–595 nm using the Varioskan LUX Multimode Microplate reader (Thermo Fisher Scientific), and absorbance values were normalised to Ethanol-treated or empty gel control wells (negative control). For Live/Dead imaging, cells were incubated with 40 µL Live/Dead stain and imaged as mentioned above (see "Cell viability and live/dead imaging" section). At the chosen endpoint, drug compounds were selected based on drug sensitivities from the original patient sample screen. Selected drug compounds were administered using the Tecan D300e digital dispenser in final concentrations ranging from 0.5 nmol/L to 5 µmol/L. Benzethonium chloride (100 µmol/L) and DMSO were administered as positive and negative controls, respectively. Seventy-two hours after drug administration, metabolic activity was measured using 40 µL of CellTiter-Glo® 3D Cell Viability Assay as described earlier (see "High-throughput drug screening" section). Raw luminescence values were normalised to control wells and calculated as viability percentage relative to DMSO wells.

### Image postprocessing and figure preparation

Images were processed using standardised microscopy workflows and best scientific practices with consistent parameter settings to ensure reproducibility and accuracy with tools described in the Reagents and Tools Table. Histology images for original patient samples were obtained from archived tissues and slides. Brightness and contrast postprocessing were applied to ensure presentability. No feature was altered, and all final images correctly represent the original data. All raw unprocessed image files are available in the source data provided for this manuscript.

BioRender.com was used to generate schematics and a graphical abstract for this publication. Figures were assembled using Adobe Illustrator.

## Quantification and statistical analysis

Statistical analysis and quantification methods are indicated in relevant "Methods" sections and figure legends. Statistical analyses were performed in GraphPad Prism software (v.10.3.0) unless otherwise stated. Data were tested for normality where applicable, and parametric (one-way ANOVA, Pearson correlation test) or non-parametric tests (Mann–Whitney U test, Kruskal–Wallis) were applied accordingly with relevant post hoc tests. Extracellular matrix gene expression analysis was performed using R (v4.2.3) and the MatrisomeAnalyseR package. Welch's t test was used for statistical evaluation of gene expression compared against the remaining core matrisome genes in the dataset. Cell viability data is presented as mean ± SD, and representative images shown from one of three independent experiments. Sample size (n) values are provided in the associated main and expanded view figures. Tumourigenicity data were analysed using a linear mixed-effects model with restricted maximum likelihood estimation (REML) with group as a fixed effect. Statistical significance levels $P > 0.05$ (ns) for not significant, $P < 0.05$ (*) for significant, $P < 0.01$ (**) for highly significant, and $P < 0.001$ (***) for very highly significant. Where exact P values are available, they are reported in the figure and corresponding figure legend.

## Data availability

This paper analyses existing patient WGS and RNAseq data, which is available upon request from ZERO Childhood Cancer Program. TSO500 and RNAseq data are deposited into a publicly available database (EGAS00001008220). Data access details are listed in the key resources table. Source data used to generate figures in the manuscript are available through EMBO Molecular Medicine BioStudies archive. Large imaging files are available through BioImage Archive (S-BIAD2130). This paper does not report original code. Any additional information required to reanalyse the data reported in this paper is available from the lead contact upon request.

The source data of this paper are collected in the following database record: biostudies:S-SCDT-10_1038-S44320-025-00152-y.

## Peer review information

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

## Acknowledgements

Children's Cancer Institute Australia is affiliated with the University of New South Wales and the Sydney Children's Hospital Network. We acknowledge ZERO Childhood Cancer personalised medicine program for providing samples and molecular data related to this project. The authors would like to thank all the patients, parents and healthcare professionals that participated in ZERO Childhood Cancer program. We would like to thank Biljana Dumevska from ZERO Core Management for her assistance in coordinating data and sample access. The authors thank the Children's Cancer Institute Animal Facility for providing support to this study. This work is supported by a National Health and Medical Research Council (NHMRC) Investigator Grants (#2016464 to MK, #1196648 to JJG), NHMRC Synergy Grant #2019056 to MK and JJG, Cancer Council New South Wales #2020797, and Cancer Australia #1185313 led by MK. We thank the Luminesce Alliance for supporting the bioinformatics analysis. We are greatly appreciative of our consumer representatives Amanda Younes, Josiane Habak, Darrin Batchelor and Simon Sleep for their support and for providing valuable feedback on the content and consumer relevance of this research. We thank Julio Ribeiro, Robert Utama, Aidan O'Mahony and the Inventia Life Science team for their technical support and advice on the 3D bioprinting. We would like to acknowledge Fei Shang and Maria Kasherman from Katharina Gaus Light Microscopy Unit at the Mark Wainwright Analytical Centre at UNSW Sydney for assistance with hydrogel cryosectioning and imaging. Anaiis Zaratzian, Andrew M. Da Silva, Michael Tayao from the Garvan Institute Histopathology Facility for the histology services. Madeleine Wheatley, Joan Solomon, Brandon Hearn and Isabella Temelkoska for technical assistance with animal experiments. Roxy Cadiz and Karina Pazaky for technical assistance with high-throughput drug screening. Michelle Henderson, Libby Huang, Luis Enriquez and Ani Lack for technical advice and assistance with PCR validation experiments. We would like to thank Claudia Flemming and Omesha Perera for their assistance in the manuscript preparation, and Anna Guller for critically reviewing the revised manuscript. Schematics for methods and results were created with BioRender.com.

## Author contributions

**MoonSun Jung**: Conceptualisation; Formal analysis; Validation; Investigation; Visualisation; Methodology; Writing—original draft; Writing—review and editing. **Valentina Poltavets**: Formal analysis; Validation; Investigation; Visualisation; Methodology; Writing—original draft; Project administration; Writing—review and editing. **Joanna N Skhinas**: Data curation; Formal analysis; Validation; Investigation; Visualisation; Methodology; Project administration; Writing—review and editing. **Gabor Tax**: Data curation; Formal analysis; Validation; Methodology. **Alvin Kamili**: Resources; Methodology. **Jinhan Xie**: Resources; Methodology. **Sarah Ghamrawi**: Investigation; Methodology. **Philipp Graber**: Formal analysis; Methodology. **Jie Mao**: Data curation; Methodology. **Marie Wong-Erasmus**: Resources; Data curation; Formal analysis; Supervision. **Louise Cui**: Resources; Data curation; Methodology. **Kathleen Kimpton**: Validation; Investigation; Methodology. **Pooja Venkat**: Data curation; Formal analysis; Methodology. **Chelsea Mayoh**: Data curation; Formal analysis; Supervision. **Angela Lin**: Resources; Data curation; Software; Formal analysis. **Emmy D G Fleuren**: Methodology. **Ashleigh M Fordham**: Methodology. **Zara Barger**: Methodology. **John Grady**: Data curation. **David M Thomas**: Supervision. **Eric Y Du**: Methodology. **Nicole S Graf**: Resources; Methodology. **Mark J Cowley**: Data curation; Supervision. **Andrew J Gifford**: Resources; Methodology; Writing—review and editing. **Jamie I Fletcher**: Resources; Methodology. **Loretta M S Lau**: Data curation; Supervision; Writing—review and editing. **M Emmy M Dolman**: Formal analysis; Supervision; Writing—original draft; Writing—review and editing. **J Justin Gooding**: Supervision; Funding acquisition; Methodology; Writing—review and editing. **Maria Kavallaris**: Conceptualisation; Resources; Funding acquisition; Writing—original draft; Project administration; Writing—review and editing.

Source data underlying figure panels in this paper may have individual authorship assigned. Where available, figure panel/source data authorship is listed in the following database record: biostudies:S-SCDT-10_1038-S44320-025-00152-y.

## Disclosure and competing interests statement

MK and JJG hold options with Inventia Life Science Pty. Ltd. MK and JJG are co-inventors of a patent (WO2017/011854) related to the bioprinting technology used in this study. JIF receives an annual payment related to the development of venetoclax from the Walter and Eliza Hall Institute's distribution of royalties scheme.

# Expanded View Figures

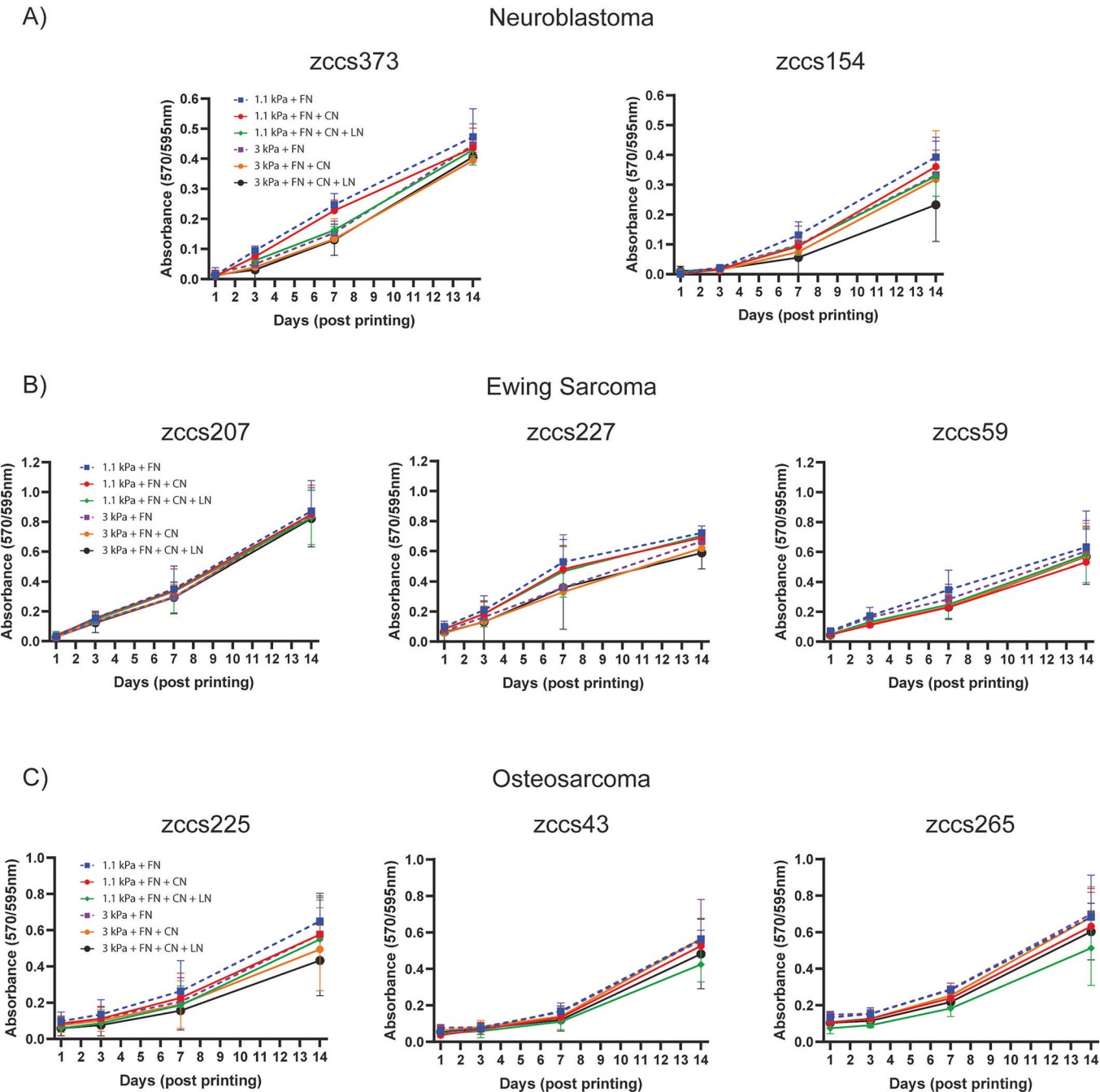

**Figure EV1. Cell proliferation of 3D bioprinted PDX tumouroids.**

Cell proliferation graphs of 3D bioprinted patient-derived PDX cells in the six hydrogel conditions. Each sample was bioprinted in either 1.1 kPa or 3 kPa hydrogels, containing fibronectin (FN) only, FN and collagen (CN), or FN, CN and laminin (LN) peptides. Proliferation rate was measured at day 1, 3, 7 and 14 post-printing. Graphs are separated into cancer types (**A**) neuroblastoma, (**B**) Ewing sarcoma and (**C**) osteosarcoma. Data are presented as mean ± SD, n.s. *P* > 0.9 (one-way ANOVA with Tukey's multiple comparisons test for each disease group). All experiments were repeated three times, except for zccs227, which had limited sample availability and is *n* = 2. Related to Fig. 2. Source data are available online for this figure.

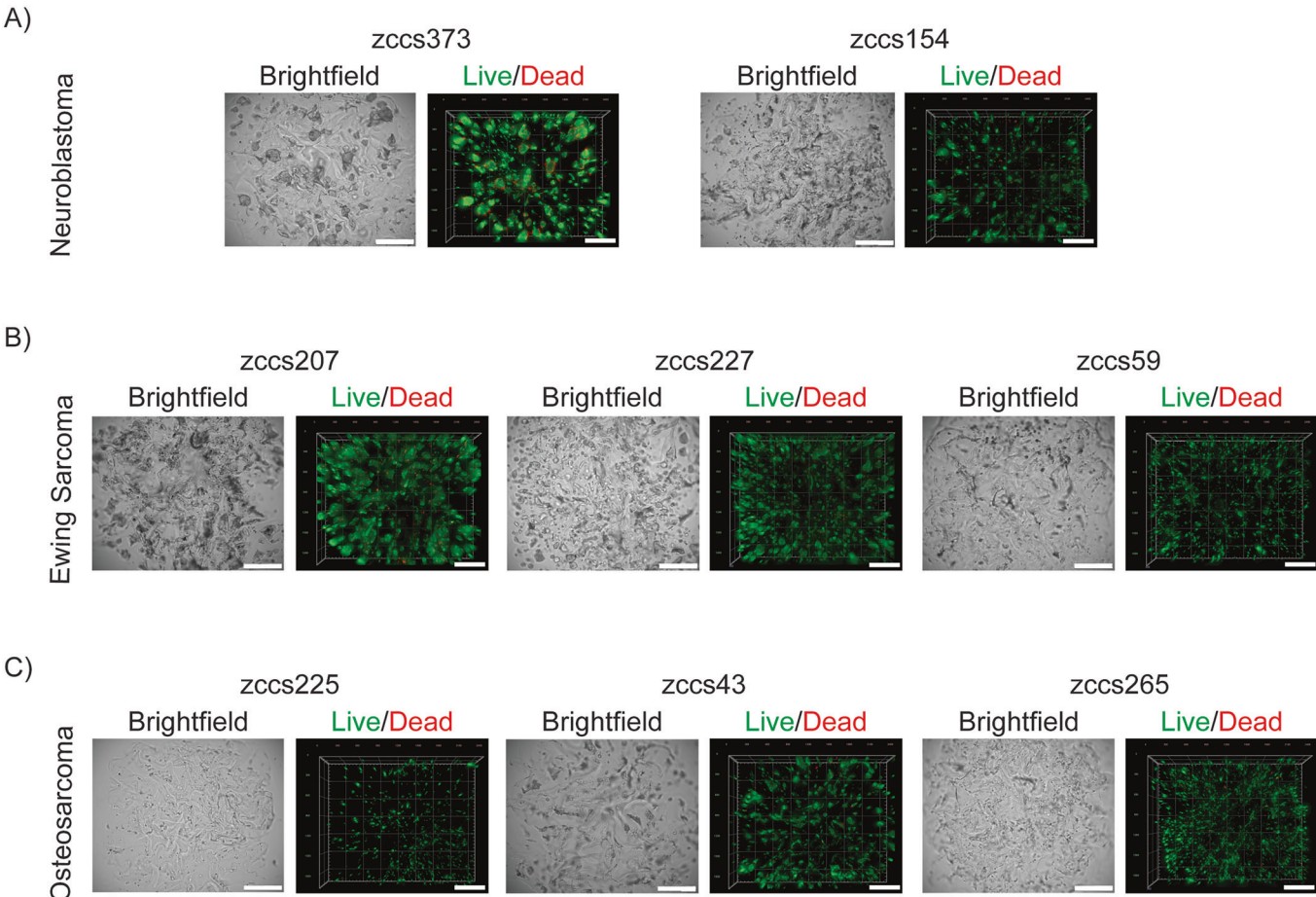

**Figure EV2. Cell viability of 3D bioprinted PDX tumouroids.**

Each sample was bioprinted in 3 kPa + FN + CN + LN hydrogels and cultured up to 14 days. Graphs are separated into cancer types (**A**) neuroblastoma, (**B**) Ewing sarcoma and (**C**) osteosarcoma. Cells were stained with calcein-AM (green; live)/ethidium homodimer 1 (red; dead) Live/Dead Assay. Z stack 3D images were taken at day 14 post-printing. Representative image shown for each sample in brightfield (left) and Live/Dead (right). Scale bars on all images are 500 μm. Related to Fig. 2.

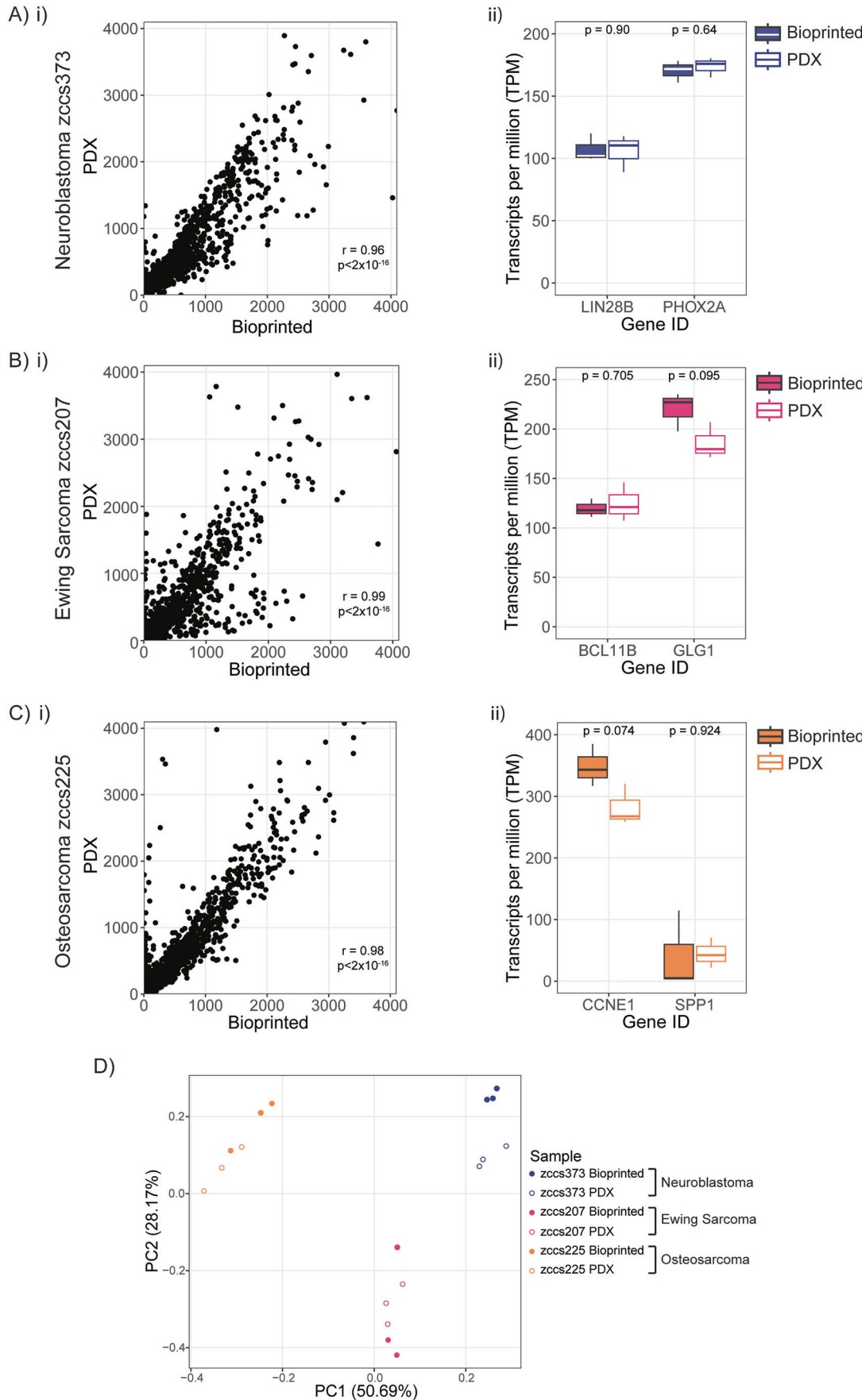

◀  **Figure EV3.  Comparative RNAseq analysis of PDX cells and bioprinted PDX tumouroids.**

(**A–Ci**) Scatterplot showing gene expression correlation between PDX and a matched bioprinted tumouroid sample for neuroblastoma zccs373, Ewing sarcoma zccs207 and osteosarcoma zccs225. Each dot represents a gene expression value pair in transcripts per million (TPM). Pearson's correlation analysis, *P* values (<2x10e-16) are indicated on corresponding plots. (**A–Cii**) Box plots showing individual transcript level (TPM) between bioprinted and PDX samples for LIN28B and PHOX2B in zcc373; BLC11B and GLG1 in zccs207 and CCNE1 and SPP1 in zccs225. Data are presented as median ±IQR; *P* values are indicated on corresponding plots. (**D**) Principal component analysis plot showing transcriptomic profiles of PDX and matched bioprinted samples across three tumour types. PC1 and PC2 account for 28.17% and 50.69% variance, respectively. Related to Fig. 3.

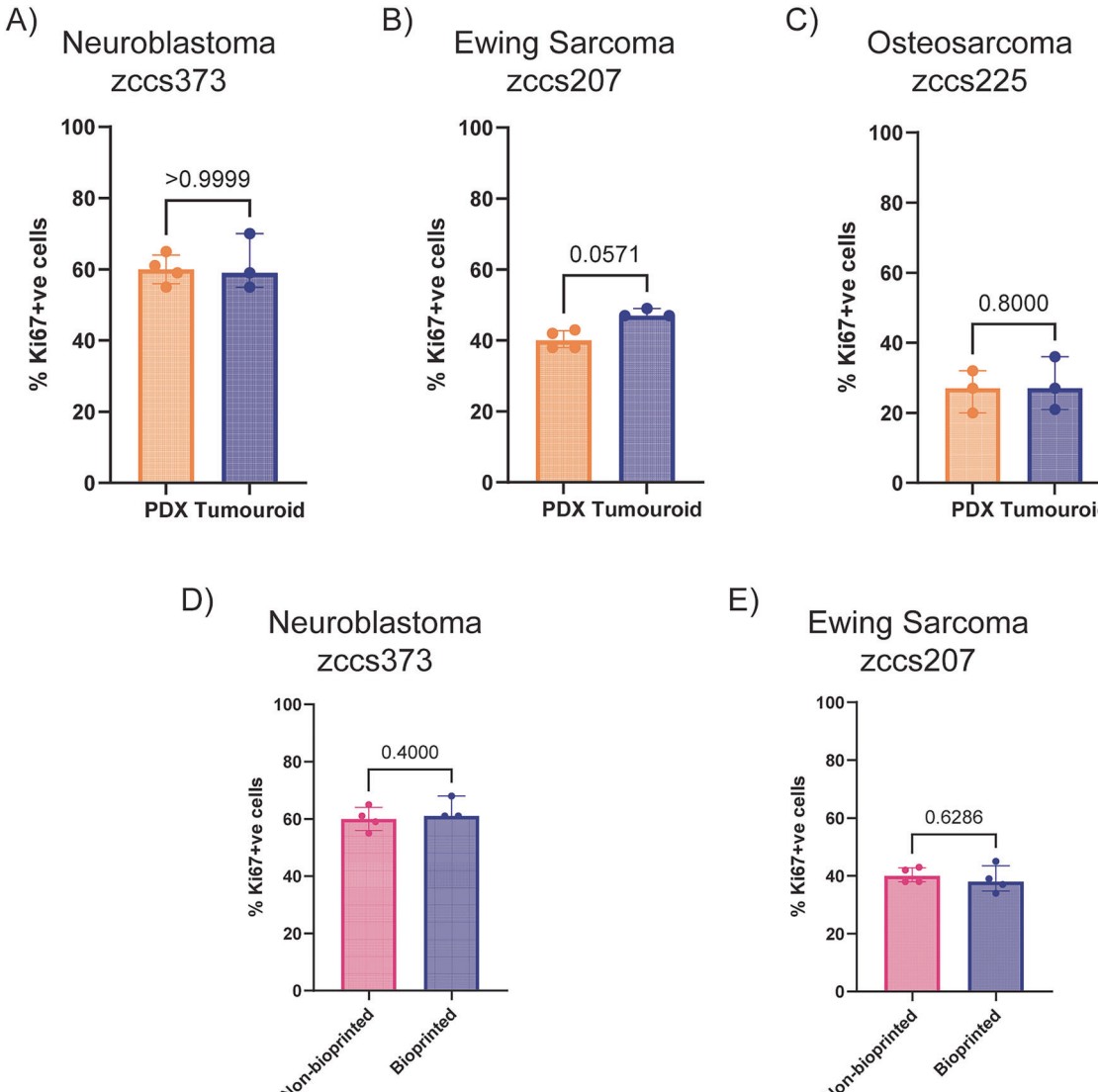

**Figure EV4. Cellular proliferation remains unchanged after the bioprinting process in vitro and in vivo.**

(A–C) Quantification of % Ki-67 positive cells in PDX tissues and bioprinted tumouroids from neuroblastoma zccs373, Ewing sarcoma zccs207 and osteosarcoma zccs225 samples. (D, E) Quantification of % Ki-67 positive cells in PDX tissues from non-bioprinted and bioprinted PDX in the tumourigenicity study. For PDX tumours, each dot represents an individual tumour (3 ROI averaged per tumour, $n = 3$–4 tumours). For PDX-derived tumouroids, each dot represents individual structures ($n = 3$). Data are presented as median ±IQR; Mann–Whitney $U$ test; Adjusted $P$ values are displayed in the figure panel. Related to Figs. 4 and 5. Source data are available online for this figure.

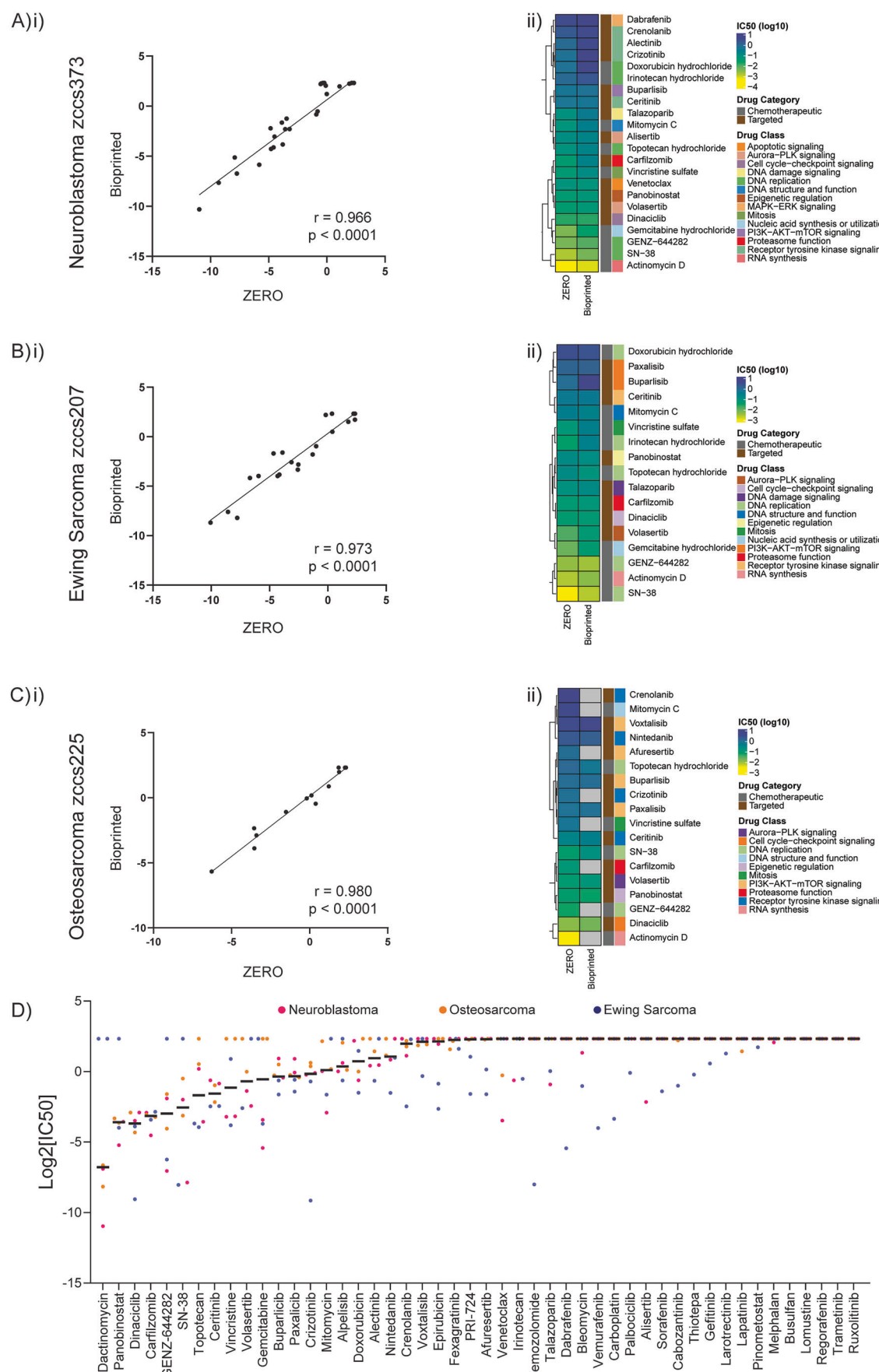

◄ **Figure EV5. Comparison of HTP drug screening approaches between preclinical testing and 3D bioprinting workflow.**

(A–C) (i) Correlation between ZERO and 3D bioprinting workflow for three PDX samples. Scatterplot with line of best fit. Each dot represents a log2[$IC_{50}$] value for a specific drug for both conditions. $r > 0.9$, Pearson's correlation analysis, $P$ values (<0.0001) are indicated on corresponding plots. (A–C) (ii) Heatmap visualisation comparing log10[IC50] values for two approaches across chemotherapeutic and targeted drug classes. $IC_{50}$ values > 5 were excluded to assist with visualisation. Grey boxes indicate unavailable $IC_{50}$ values for zccs225 due to technical limitations at the time of the experiment. (D) Log2[$IC_{50}$] distribution across all samples for a 48-drug library. Drugs were ordered based on the lowest to highest median Log2[$IC_{50}$] values, followed by lowest quartile and then lowest detected Log2[$IC_{50}$] values. Related to Fig. 6. Source data are available online for this figure.

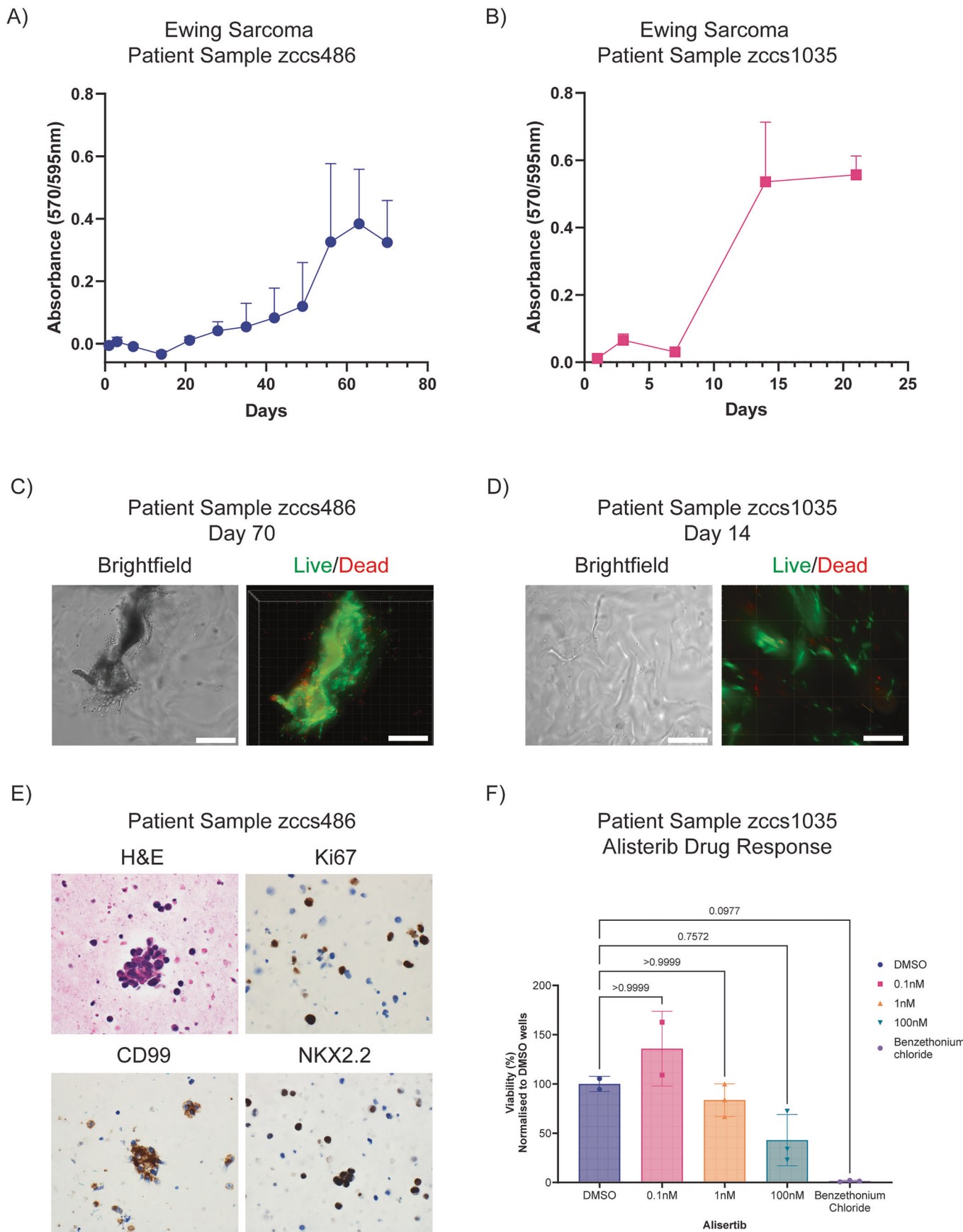

**Figure EV6.  Compatibility of original patient samples with 3D bioprinting for sample expansion and drug screening.**

(A, B) Cell proliferation of 3D bioprinted patient cells for zccs486 and zccs1035. Data are presented as mean ± SD. (C, D) Representative brightfield (left) and live/dead images (right) for 3D bioprinted patient tumouroids at day 70 (zccs486) and day 14 (zccs1035). Scale bars on all images are 200 µm. (E) Histology and immunohistochemistry images for zccs486 sample. Panels H&E, Ki-67, Ewing sarcoma tumour-specific markers CD99 and NKX2.2. (F) Alisertib drug response in 3D bioprinted patient tumouroids, Kruskal–Wallis with Dunn's Test; n.s. *P* values are indicated on corresponding plot. Source data are available online for this figure.

