## [Peer Review File · Molecular Systems Biology]

High-throughput 3D engineered paediatric tumour models for precision medicine.

MoonSun Jung, Valentina Poltavets, Joanna Skhinas, Gabor Tax, Alvin Kamili, Jinhan Xie, Sarah Ghamrawi, Philipp Graber, Jie Mao, Marie Wong-Erasmus, Louise Cui, Kathleen Kimpton, Pooja Venkat, Chelsea Mayoh, Angela Lin, Emmy Fleuren, Ashleigh Fordham, Zara Barger, John Grady, David Thomas, Eric Du, Nicole Graf, Mark Cowley, Andrew Gifford, Jamie Fletcher, Loretta Lau, M. Dolman, John Gooding, and Maria Kavallaris

Corresponding author(s): Maria Kavallaris (M.Kavallaris@ccia.unsw.edu.au) , John Gooding (justing.gooding@unsw.edu.au)

Review Timeline:

Transfer Date:	13th Aug 25
Editorial Decision:	13th Aug 25
Revision Received:	2nd Sep 25
Accepted:	9th Sep 25

Editor: Jingyi Hou

Transaction Report: This manuscript was transferred to Molecular Systems Biology following peer review at EMBO Molecular Medicine.

Black text: Reviewer Comments

Blue text: Author responses

We are grateful to all the reviewers for their time in critically reviewing our paper, support of the study, insightful comments, and suggestions.

R1

Referee #1 (Comments on Novelty/Model System for Author):

It should be noted that many figures in the manuscript lack essential statistical analysis. The original publication describing this 3D-printing technology appeared in a different journal. Most critically, the authors did not include any data directly derived from patient samples, which is essential to substantiate the claims they are making.

The comments are covered under specific questions from the reviewer below.

Referee #1 (Remarks for Author):

In this manuscript, MoonSun Jung et al. report a high-throughput 3D bioprinting platform designed to generate patient-derived neuroblastoma and sarcoma organoids within tunable ECM-mimicking hydrogels. The authors claim that this platform successfully preserves the genetic and phenotypic characteristics of neuroblastoma and sarcoma samples and can be adapted for high-throughput drug screening workflows. This approach shows potential to address some existing challenges in pediatric tumor modeling, particularly regarding the speed and scalability for drug testing. However, to confirm the robustness and translational value of this platform as a tool for personalized treatment strategies in pediatric cancers, the authors should validate it with primary patient tissue samples and address some concerns.

Each point has been addressed in detail below.

Major Comments:

1. While the proof-of-concept results presented are promising, validation using primary patient tissue samples is necessary to fully support the authors' claims. The authors should further characterize organoid culture success rates and describe the characteristics observed in patient-derived samples.

We appreciate the reviewer's comment regarding the need to validate our proof-of-concept approach in 3D bioprinting workflow using primary patient tissue samples. As noted in the manuscript, a key challenge in the field for functional precision medicine is the limited sample sizes available from paediatric patients (most often a biopsy). Genomic analysis is prioritised in our ZERO precision medicine program often resulting in limited samples. To cover the wider spectrum of high-risk paediatric solid cancers and to develop a reliable platform, we obtained our proof-of-concept using patient-derived cells from PDXs.

As per the reviewer's recommendation, we have now obtained Ewing Sarcoma primary material from patients. These samples had been cryopreserved directly from the original tumour biopsy. Following thawing of the samples (zccs1035 and zccs486), we successfully bioprinted and grew them in our system (**new Supplementary Figure 6**). As we had low numbers of patient cells in each sample we

prioritised different assays for each sample. For zccs486 we validated that the pathological markers from the tumour used for diagnosis were still expressed by IHC. For sample zccs1035, we obtained dose response data for Alisertib. Due to the majority of samples obtained being biopsies, the ZERO Childhood Cancer program has limited viably cryopreserve tumour cells. Our preliminary results provided initial confirmation that the 3D bioprinting approach is feasible for culturing and expansion of the primary patient tumour cells. This is crucial as we previously showed that the expansion of primary patient cells in vitro remains difficult for paediatric cancers (**Supplementary Table 2**).

As a result of our study, we are working with the ZERO team (16 co-authors on this paper) for our platform to be included in the ZERO2 clinical study where we will focus on obtaining a larger patient number of primary patient samples and align this with clinical outcomes of drug recommendations.

Our preliminary data on patient samples confirms the ability to bioprint, expand and screen patient tumour cells directly on patient samples. Changes and results are described in the following sections:

- Supplementary Figure 6
- Supplementary Table 1
- Results (p.15, 354):

“Next, we determined the feasibility of 3D bioprinting to expand original patient tumour samples for characterisation and drug screening. We accessed two cryopreserved high-risk Ewing sarcoma (EWS) patient samples - zccs1035 and zccs486 (Table S1). Patient cells were bioprinted as described for Fig. 6. Initially, resazurin-based assays demonstrated low metabolic activity in both samples, followed by a marked increase in activity between days 7–14 for zccs1035 (Fig. S6A) and days 50–60 for zccs486 (Fig. S6D). Live/dead imaging confirmed viability during culture for both samples (Fig. S6B, E). Histological analysis of the zccs486 cells confirmed that the cells expanded post-bioprinting were proliferating (Ki67) and expressed key EWS disease-specific markers, including CD99 and NKX2.2 (Fig. S6F).

Due to limited sample availability, we conducted a single drug screen experiment on Ewing Sarcoma zccs1035 using three concentrations of alisertib, alongside negative (DMSO) and positive (100 μ M Benzethonium Chloride) controls. This data demonstrates the ability of the 3D bioprinted and expanded direct patient sample to respond to alisertib showing a trend towards decrease in viability at 100nM of drug compared to vehicle control. While the small sample size and variability limits the robustness of these findings, they provide preliminary evidence that direct bioprinting of patient samples for drug testing is feasible using our 3D bioprinting workflow.”

- Discussion (p.19, 443):

“Our study demonstrates the feasibility of generating reproducible 3D ECM-mimic models for direct high-throughput drug screening bypassing conventional cell expansion approaches, aiming to preserve the molecular and phenotypic fidelity of patient-derived samples with potential clinical translatability. A limitation has been the small number of direct patient tumour samples we were able to include in our study. Future studies will link to a clinical trial to obtain additional patient samples to expand on this work.”

2. Beyond the histological analysis, the authors should include molecular comparisons of 3D bioprinted organoid cultures with corresponding xenograft tissues and, if available, the original patient tissues. Bulk RNAseq or scRNAseq data would provide more robust evidence that these bioprinted tumor organoids replicate the molecular characteristics of the original tumors.

To provide robust molecular comparison between 3D bioprinted tumouroids and corresponding xenograft tissues, in addition to the previously presented data (**Figure 3, Supplementary Figure 4**), we performed RNA-seq on patient-derived cells from three representative PDXs, both pre- and post-3D bioprinting (**new Supplementary Figure 3**). It is clear from our RNAseq data that the bioprinted tumouroids replicate the molecular expression profiles of xenograft tumours, reinforcing our model's ability to retain original molecular characteristics after bioprinting.

The following text has been added to Results section (p.10, line 235):

“To further investigate the effect of bioprinting and culture on the molecular characteristics of the 3D tumouroids, we performed a comparative RNA-seq analysis on patient-derived cells from three representative PDXs, both pre- and post-3D bioprinting (Supplementary Figure 3). Bioprinted tumouroids exhibited a strong positive correlation in gene expression profiles with patient-derived xenograft (PDX) cells across neuroblastoma (zccs373, $r=0.96$, $p<0.001$), Ewing sarcoma (zccs207, $r=0.99$, $p<0.001$), and osteosarcoma (zccs225, $r=0.98$, $p<0.001$) samples (S3Ai-Ci), confirm high molecular fidelity post bioprinting. We further analysed PDX and bioprinted samples for expression levels of disease-specific markers used in prognosis and diagnosis by the ZERO Personalised Medicine Team¹. In neuroblastoma (zccs373), LIN28B and PHOX2A showed comparable expression levels (S3Aii). Similarly, BCL11B and GLG1 were assessed in Ewing sarcoma (zccs207), and CCNE1 and SPP1 in osteosarcoma (zccs225) (S3Aii-Cii). These findings indicate that key disease-specific molecular characteristics are preserved following 3D bioprinting. We next performed principal component analysis (PCA) which demonstrated that PDX and their corresponding bioprinted samples clustered closely, indicating a high degree of similarity in global gene expression profiles (S3D). Furthermore, molecular aberrations previously identified by targeted panel sequencing (TSO500) were confirmed by RNAseq (Supplementary Table 5). Our findings confirm that bioprinted tumouroids faithfully replicate the molecular expression profiles of xenograft tumours, reinforcing our model's ability to retain original molecular characteristics after bioprinting.”

3. To substantiate the platform's suitability for high-throughput drug screening, the authors should report the Z-prime value of the drug assays. This metric would help demonstrate the robustness and reliability of drug screening with bioprinted organoid cultures.

Thank you for the suggestion. As requested, we have included both Pearson correlation coefficient and Z'-factor calculations in the Supplementary Materials (**new Supplementary Table 8**).

The workflow for the drug screening platform applied to 3D bioprinted samples has been previously published² and is described in detail in the Methods section (page 30, 775). To further validate the suitability of patient-derived 3D bioprinted tumoroids for high-throughput screening (HTS), we have incorporated established quality control (QC) criteria details into the Methods section (page 31, 801):

“Quality control cut offs were defined based on two key metrics: (1) a Pearson correlation coefficient > 0.7 between technical replicate plates, and (2) a robust Z'-factor >0.3 for both replicates.”

A robust Z-prime was calculated based on the following formula:

$$Z'_{robust} = 1 - \frac{3 \times (MAD_{positive} + MAD_{negative})}{|Median_{positive} - Median_{negative}|}$$

MAD is a Median Absolute Deviation

4. The authors claim that batch-to-batch variability is minimized with the use of ECM-mimicking hydrogels. To support this, they should conduct experiments with different ECM batches, assessing growth rates and drug responses, and demonstrate that this ECM performs better than Matrigel and other animal-derived products in terms of consistency and functionality.

This is beyond the scope of our paper. Here we employed commercially available, quality-controlled bioinks with defined peptide content that are specifically designed for use in bioprinting applications. A direct comparison with Matrigel is not appropriate in this context, as Matrigel is well known for its batch to batch variability - a limitation that has been well-documented in the literature (reviewed in Aisenbrey, E.A., and Murphy, W.L. (2020). Synthetic alternatives to Matrigel. *Nat Rev Mater* 5, 539-551.) In addition, Matrigel is not compatible with our bioprinter.

To highlight the advantages of defined synthetic hydrogels over biologically derived matrices such as Matrigel and collagen, we cited relevant literature within the manuscript (p.5, paragraph 2; p.16, 390-392; p.18, 432-437) . Notably, defined hydrogels offer greater reproducibility and tunability, which are critical for standardised drug screening workflows. Furthermore, our system allows for the efficient dissociation and recovery of viable tumour cells from the hydrogel matrix, enabling downstream molecular analyses such as RNA sequencing, qPCR, and protein profiling—procedures that are significantly more challenging when using Matrigel.

5. In Figure 4, PDX IHC data (e.g., Ki-67, PHOX2B, CD99 and SATB2) should also be included to compare PDX and bioprinted tumor organoids. Additionally, please provide a quantitative graph comparing the values between these two groups. Similar quantitative analysis for the IHC staining is also needed for Figure 5.

Addressing the reviewers' helpful suggestion, we have now included a side-by-side histological comparison of the original patient tumour samples, PDX tissues, and 3D bioprinted tumoroids from PDX cells (**Figure 4**). This further highlights the compatibility of the bioprinted cells to maintain expression of tumour specific markers.

Regarding quantification of IHC staining, it is important to note that PDX tissues and hydrogel-embedded tumouroids were processed using different methods - formalin-fixed paraffin embedding (FFPE) and hydrogel cryopreservation, respectively - and harvested at different time points. As advised by our two pathologists on this paper, these differences can affect the comparison of IHC staining across samples. Also, as is well established in the field, IHC for diagnostic pathology is primarily a semi-quantitative technique, used to assess the presence or absence of specific markers. Nonetheless, we have included qualitative images of proliferation marker expression and quantified the percentage of Ki-67 positive cells relative to the total number of cells. This Ki-67 proliferation index is presented in **Supplementary Figure 4**, corresponding to the data shown in **Figure 4** (PDX tissues and hydrogel-embedded tumouroids) and **Figure 5** (PDX tissues from cells engrafted pre- and post-bioprinting).

The text has been modified and the following added to Results section (p.11, 253):

“Next, the morphologic features of the 3D bioprinted PDX tumouroids were characterised in comparison with the features of the patient-derived xenografts (PDXs) and the original patient

samples. PDX cells were bioprinted and cultured for 7 to 14 days, prior to preparation and cryosectioning. Samples were stained with both routine H&E and tumour-specific immunohistochemical staining. The bioprinted tumouroids recapitulated the morphologic features of the original patient sample and PDX from which they were derived (Fig. 4A-C)."

(p.12, 267):

"Quantification of Ki67 expression showed comparable levels between PDX tissues and 3D bioprinted tumouroids indicating similar levels of proliferative activity in the cells post bioprinting (Fig. S4A-C). These results confirm that the morphologic and diagnostic immunohistochemical characteristics of both original patient samples and PDXs are preserved in 3D bioprinted tumouroids."

Minor Comments:

6. There is insufficient detail regarding cell densities and specific culture conditions in the 3D-printed organoid cultures. While the authors mention total cells per well, they should also specify the volume of ECM used per well in bioprinting, as well as the media formulation utilized in the cultures.

Details of the cell densities, media formulations and specific culture conditions are described in the originally submitted manuscript:

Cell densities – p.21, 509, 512, 514-515

Media formulations – p. 20, 495-500

Specific culture conditions - p. 22, 523

The concentration of the ECM-mimic peptides and bioprinting droplet volume are proprietary RASTRUM™ Inventia information and could not be reported in the manuscript. To address the reviewer request regarding cell densities, we included an extended description of the bioprinting process and detailed information on cell concentrations used in the Methods section *Bioprinting* p.24, 551:

"The RASTRUM™ 3D bioprinter utilises two elements: bioink and activator. Cells are resuspended in activator solution and loaded into the bioprinter, along with bioink, in separate reservoirs within the printer cartridge. Both solutions are bioprinted and form an instant gel when combined in a microplate. Printing protocols and hydrogel design were generated using RASTRUM™ Cloud (Inventia Life Science). In this study we utilised four bioprinting models: Imaging Model (25×10^6 cells/mL; ~8000 cells printed per well), Immunohistochemistry Model (25×10^6 cells/mL, ~16000 cells printed per well), Large Plug Model (12.5×10^6 cells/mL; ~36000 cells printed per well); HTP Model ($2.04-4.08 \times 10^6$ cells/mL; ~2000-4000 cells printed per well) These seeding densities were used for all patient-derived samples except for zccs227, where the concentration of cells per ml was doubled for all bioprinting models."

7. Most figures lack statistical analysis, and there is limited information on the number of replicates for each experiment.

We have now included statistical analysis and sample number to the following Figures and corresponding legends:

Figure 1

Figure 2

Figure 5

Supplementary Figure 1

New Supplementary Figure 3

Supplementary Figure 4

Supplementary Figure 5

Supplementary Figure 6

We also provided extended information on statistical analysis in the Methods section p.33, 772

In the original manuscript, Figure descriptions provide sample number of replicates for each experiment:

Figure 1

Figure 2

Figure 5

Supplementary Figure 1

8. It would be more effective to present a single plot for each sample comparing the growth rates of organoid cultures under different hydrogel conditions (FN, FN + CN, FN + CN + LN). Similarly, to highlight the function of 1.1kPa+FN+CN+LN ECM, the results of 3 kPa+FN+CN+LN ECM should be provided in the same figure.

We have addressed the reviewer suggestion and modified **Supplementary Figure 1** to reflect all the conditions tested per sample in a single plot.

9. The authors should discuss both the advantages and limitations of this 3D bioprinting method in comparison to other advanced organoid-based techniques, citing relevant state-of-the-art studies to provide broader context for their findings.

To address this request, we included the following paragraph in the discussion (p.18, 432):

“Our research demonstrates the application of the 3D bioprinting technology in creating patient-derived cancer models for downstream preclinical testing. Current advanced methodologies for preclinical 3D culture testing include scaffold-free spheroids²⁻⁴ and Matrigel or basement membrane extract (BME) cultures^{5,6}. These systems replicate key tumour features yet have critical limitations – (i). manually labour intensive; (ii). either lack or have undefined ECM

composition; (iii). suffer limited reproducibility⁷. Our 3D bioprinting approach provides a standardised and high-throughput method for patient-derived tumour modelling. It allows scalable production of mini-tumour models in minutes in standardised ECM-mimic hydrogel with defined mechanical and biochemical properties. Bioprinting enables precise control of size, structure, and cell distribution. Limitations include reliance on proprietary formulations of bioinks and limited incorporation of specialised extracellular matrix components or non-standard additives.”

10. The original publication describing this 3D-printing technology appeared in a different journal

Regarding the “*original publication describing this 3D-printing technology appeared in a different journal*” we have checked the references and these are correct. The original bioprinter was described in Utama RH, et al. *A 3D bioprinter specifically designed for the high-throughput production of matrix-embedded multicellular spheroids. iScience 23(10):101621 (2020)*⁸. The tunable bioinks and new printing method (encapsulated cells) that was used in the paper cited Utama RH, et al. *A Covalently Crosslinked Ink for Multimaterials Drop-on-Demand 3D Bioprinting of 3D Cell Cultures. Macromol Biosci., 21(9):e2100125 (2021)*⁹.

Unlike what is presented in our current manuscript, none of the above papers developed a platform to investigate and validate patient derived xenograft cells, high-throughput drug screening and patient tumour cells.

R2

Referee #2 (Remarks for Author):

In the manuscript "Engineered paediatric tumours retain patient tumour genotype and phenotype for precision medicine" the authors generated 3D bioprinted co- culture system with patient cancer cells and hydrogels. The experiments are well- performed and novel techniques have been developed. Furthermore, the authors have been able to generate new models of human pediatric neuroblastoma and sarcomas.

We thank the reviewer for their positive feedback and their valuable suggestions.

Main Concern:

- 1. I would not use the word "organoids" for the 3D bioprinted model. Organoids resemble (in a limited way) human organs, with several cell types and 3D organization with specific cells in specific locations. In their model, only cancer cells are present, therefore can not be called organoids.**

We thank the reviewer for this suggestion. In the previous version of the manuscript, we used the term "tumour organoids," which is acceptable for patient-derived tumour cells e.g. patient-derived tumour organoids¹⁰ (PDTOs). However, for clarity and consistency, we have now adopted the term "tumouroids" throughout the text.

- 2. The 3D bioprinted model is very complex and well-developed. The authors suggest using their new model as a valuable model for drug testing. Since the new model is time-**

consuming and expensive to implement, the authors should show a real benefit in using their model. At least, the authors must show that their 3D bioprinted models are better in drug response than 2D cell culture systems.

We appreciate the reviewer comment. Indeed, our proof-of-concept showing that our bioprinted tumouroids are representative of high-risk childhood cancer models, can be rapidly implemented, and serve as valuable models for direct in situ high-throughput drug testing. As discussed in the manuscript, the key reasons for not using 2D models, as we and others have previously shown (**Supplementary Table 2**), is that patient tumour cells do not readily grow directly in conventional cultures, are not representative of how tumour cells grow and often require in vivo expansion as PDXs to obtain sufficient sample for drug screening – with PDX success for individual samples being variable and time to expansion ranging between 57-179 days depending on the sample (**Supplementary Table 3**).

Nevertheless, to address this question and to provide a relevant benchmark, we were able to compare our 3D bioprinting approach with the results of our existing preclinical testing protocol (ZERO) in three PDX tumour samples. Consistent with previous studies in 2D vs 3D (eg. ¹¹⁻¹³), a difference across three HTP screens was that 3D bioprinted cultures exhibited reduced sensitivity, with higher IC₅₀ values compared to ZERO. This suggests that lower drug concentrations were required to achieve 50% cell killing in the ZERO protocol compared to 3D bioprinted samples (**Supplementary Table 7**).

The following edits have been made to the manuscript (p. 13, 295):

*“To initially assess the applicability of our 3D bioprinting approach for preclinical drug testing, we performed a side-by-side comparison of HTP drug screening using patient-derived xenografts cells, applying either the standard ZERO protocol² (2D or 3D spheroids in vitro culture) or the 3D bioprinting approach. Our findings demonstrate a high concordance in drug sensitivity profiles between the 3D bioprinted and ZERO preclinical testing protocols (**Supplementary Figure 5**). Both approaches reliably identified drug responses, with strong correlation observed between the two approaches—for example, $r = 0.96$ for neuroblastoma zccs373, $r = 0.97$ for Ewing sarcoma zccs207, and $r = 0.98$ for osteosarcoma zccs225 samples (all $p < 0.001$) (**S5 Ai-Ci**). Heatmap clustering further confirmed similar sensitivity patterns across drug classes, including both chemotherapeutics and targeted agents (**S5 Aii-Cii**). However, a consistent difference across three HTP screens was that 3D bioprinted cultures exhibited reduced sensitivity, with higher IC₅₀ values compared to ZERO (**Supplementary Table 7**). This suggests that lower drug concentrations were required to achieve 50% cell killing in the ZERO protocol compared to 3D bioprinted samples.”*

3. In Figure 4 the authors compared 3D bioprinted model with patient-derived xenograft. The authors should show staining for the patient-derived xenograft. It is also required to show patient samples to fully validate their new model.

We have now addressed this question in response to Reviewer 1 Question 5 and modified **Figure 4** and included **Supplementary Figure 4**.

R3

This study by Jung et al reports on the establishment of PDX-derived tumor organoids bioprinted in ECM with the intent to perform precision medicine screenings. The authors include 2 neuroblastomas, 3 ewings and 3 osteosarcomas which have been obtained from PDX samples. The paper is not immediately clear on this aspect (patient tumor vs. PDX), but it should be clarified early on that this work is performed on PDXOs rather than PDOs, and unambiguously represented within the whole paper. While the study is generally robust, given the small sample size and published literature on bioprinted organoids, PDOs for rare pediatric cancers and HTS including correlation to clinical outcomes, the impact is considered limited.

We would like to thank the reviewer for thorough and detailed review of the manuscript. The points raised are addressed individually in the sections that follow.

We acknowledge that the term "patient-derived cells" may have caused confusion; therefore, we have clarified this terminology in the Introduction on p. 6 and included explicit terminology throughout this manuscript.

Some major points below:

1. *"For example, osteosarcomas are difficult to grow ex vivo, often requiring in vivo engraftment to expand cell numbers for drug screening"*

This is inaccurate, see Peterziel et al, npj Precision Oncology 2022 and Al Shihabi et al, Cell Stem Cell 2024. Osteosarcoma, including from biopsies, are very amenable to direct culturing in ECM to establish patient-derived organoids as has been published and well documented before. Moreover, Peterziel et al also included pediatric brain tumors while Al Shihabi et al included both pediatric and adult sarcomas organoids, but with a large proportion of osteosarcoma and other pediatric sarcomas. These studies should be appropriately mentioned.

Our study differs from those cited by the reviewer. The above studies demonstrated the feasibility of culturing patient-derived spheroids or Matrigel-embedded patient-derived tumoroids for drug testing applications. However, both studies (Peterziel et al, npj Precision Oncology 2022 and Al Shihabi et al, Cell Stem Cell 2024.) describe culture prior to the drug screening - for instance, Peterziel describes the expansion of patient-derived cells in culture for fewer than seven days prior to drug testing on 3D spheroids, notably without the inclusion of an ECM-like matrix. In contrast, Al Shihabi reports short-term culture (3–5 days) using Matrigel to support the growth of patient-derived cells.

Regarding the success rates of the expansion of osteosarcoma patient samples, Peterziel et al. do not provide specific success rates for paediatric osteosarcoma ex vivo cultures. Based on two-year pilot data from the INFORM paediatric precision oncology program, osteosarcoma samples represented approximately 11% of all tumour specimens subjected to ex vivo culture and drug sensitivity profiling (DSP). Across the full cohort (n = 132), 67% (89/132) of samples underwent either full or partial drug screening, with 78% (69/89) of those passing internal quality control. Specifically, of the 17 osteosarcoma samples, 7 (41.2%) underwent full screening, 2 (11.8%) had partial screening, and 8 (47.1%) either failed quality control or did not proceed to screening.

Similarly, Al Shihabi et al. do not report separate success rates for adult or paediatric osteosarcoma-derived organoid expansion. Instead, they report an overall organoid establishment success rate of 93% across all sarcoma samples tested ex vivo between 2018 and 2022. In contrast, the ZERO Childhood Cancer Precision Medicine Program reported a significantly lower success rate of 13% for direct osteosarcoma cultures, based on 23 attempts using either fresh or cryopreserved material (**Supplementary Table 2**).

We have now modified the sentence to include abovementioned studies as follows (p.5, 100):

"For example, paediatric osteosarcomas are challenging to grow ex vivo, often requiring either ex vivo expansion or in vivo engraftment to obtain cell numbers required for comprehensive personalised drug screening ³. Recent studies have had some success with short-term culturing of osteosarcomas using Matrigel ⁶.

2. *"In this proof-of-concept study, we address some key limitations in precision medicine, namely the ability to maintain and expand difficult to grow freshly isolated patient-derived cells in a HTP fashion and conduct robust HTP drug screening in an environment that mimics tumour growth in a timely manner."*

Notwithstanding previous studies that have performed screenings in pediatric patient-derived samples in similar if not shorter timelines, the point of addressing the issue to maintain and expand difficult to grow freshly isolated patient-derived cells is moot given that the authors are using abundant, optimally proliferative PDX-derived cells.

This has been addressed in our response to Reviewer 1 Q1. It should be noted that cells derived fresh from PDX cultures are challenging to expand in 2D cultures and our approach was valid to establish the platform. To make it clear that the majority of the study was performed on patient-derived cells obtained from PDXs, we have now modified the statement below as follows:

*"In this proof-of-concept study **using cells from patient-derived xenograft models**, we address some key limitations in precision medicine, namely the ability to maintain and expand difficult to grow freshly isolated patient-derived cells in a HTP fashion and conduct robust HTP drug screening in an environment that mimics tumour growth in a timely manner" (p.6, 122)*

3. *"Thus, our analysis identified key genes that encode for structural components within childhood neuroblastoma and sarcoma tumours."*

Proteomics is the method of choice to determine ECM components, including abundance and critical PTM, see Shao et al, Nucleic Acid Research 2020. Limited gene expression-protein abundance correlation, particularly for secreted, low-turnover proteins and the inability to determined modifications that are critical to function limit the utility of RNA-seq approaches in this context. As such, the limitations of the approach chosen should be properly acknowledged.

We agree that detailed protein data would be valuable for identifying extracellular matrix (ECM) components within the tumour microenvironment. However, a systematic proteomic analysis of neuroblastoma and sarcoma ECM falls beyond the scope of the current study.

Patient sample availability for ECM proteomics was not feasible with the limited material available. Therefore, with full understanding and appreciation of the potential RNA-protein discordance, we used the RNAseq database from the ZERO program as the first-line information source to characterise the ECM of the studied paediatric solid tumours.

We have now addressed this in the manuscript with the following statement:

“While gene expression analysis provides valuable insights into ECM gene expression, it may not directly reflect protein abundance or post-translational modifications that mediate ECM function. Many ECM proteins undergo complex processing and structural assembly that cannot be inferred from transcriptomic data alone. Further studies are required address ECM protein expression in patient tumour samples.” (p.7, 145).

4. "Post-bioprinting, patient-derived neuroblastoma (Figure 2Ai), Ewing sarcoma (Figure 2Bi) and osteosarcoma (Figure 2Ci) cells embedded in the tripeptide hydrogel (FN + CN+ LN) exhibited similar growth dynamics, characterized by consistent proliferative activity up to day 14."

A critical comparison here would be overall viability and gene expression immediately prior and immediately post-bioprinting, to quantify whether the printing process itself can alter clinical sample behavior (or composition). This is a critical control which is missing.

Our RNAseq analysis of PDX cells before and after bioprinting confirmed that the global gene expression of the cells was not affected by bioprinting process (**Supplementary Figure 3**).

5. "Collectively our data suggest that neither the stiffness, nor specific peptide combinations had a major impact on cellular proliferation across all disease types"

That no difference is observed in these matrices is perhaps unsurprising, but also calls for a comparison with widely used products such as cultrex or Matrigel - how does growth compares with the standard ECMs used in the field? A quantifiable and statistically significant difference would allow the reader to evaluate whether these engineered matrices are not only as good but potentially superior to commercial products and provide important context.

This has been addressed in response to Reviewer 1, question 4.

6. "This hydrogel was selected for subsequent experiments as it supports the cellular growth of high-risk paediatric tumour cells across multiple cancer types"

The rationale should be clarified, as the authors own data suggests that both compositions tested are equally as good at supporting growth of such tumors.

In the manuscript we indicated that one hydrogel combination was selected based on

- i.) Matrisome analysis (**Figure 1**) – fibronectin and collagen I (p.7, 162)
- ii.) Importance of laminin as a key basement membrane component for integrin-mediated adhesion (p.8, 182)

- iii.) No difference between 1.1 kPa and 3 kPa condition – 3/8 samples tested were relapse tumours from lung metastasis and 1/8 samples had isolated lung metastasis.

To clarify for the reviewer and readers we have now included the following statement:

"This one type of hydrogel was selected for subsequent experiments as it supports the cellular growth of high-risk paediatric tumour cells across multiple cancer types and specifically given that many samples in our cohort were from relapsed metastatic lesions to the lung the 1.1kPa stiffness was most appropriate" (p.8, 185)

- 7. Figure 3D needs to include both the patient tumor and organoid plots to allow the reader to compare. A trio of patient-PDX-PDO would be particularly helpful. The data suggests that structural variants are missing in the organoid samples: if these are validated by other mean and present, the figure should reflect that. If for some reason the SVs are undetected with the current approach and subsequent validation, and no other type of sequencing can be performed, then it should be clearly reported.**

Figure 3D – we are not able to include Circos plots for bioprinted samples alongside patient samples as the WGS sequencing was not performed on those samples. We included a comparative analysis of whole genome sequencing (WGS) and RNA sequencing (RNA-seq) data from original clinical patient tumours alongside data from 3D bioprinted patient-derived xenograft (PDX) cells, analysed using the Illumina TruSight Oncology 500 (TSO500) panel (**Figure 3B**).

In the original manuscript, we reported the absence of detectable structural variants (SVs) in two samples, zccs43 and zccs265. We attributed this to the inherent limitations of targeted sequencing approaches, such as TSO500, in identifying structural variants (see p. 10, lines 227-228). Additionally, the TSO500 panel does not include the LSMD1 gene, which is why there is a lack of SV detection in this case. To address this limitation, we performed subsequent validation using amplicon-based analysis (**Figure 3D**).

We included the following clarification:

"Original zccs43 patient sample contained intragenic structural variant of TP53 with breakpoint in the noncoding region (exon1), and zccs265 patient sample included intergenic TP53- LSMD1 structural variant. The presence of these structural variants was not detected in corresponding PDX-derived tumouroids using targeted panel sequencing due to the technical limitations of the assay. Nevertheless, we subsequently confirmed the genetic match of the PDX-derived tumoroid to the original patient tumour using amplicon analysis (p.10,215).

- 8. The authors observed an increase in tumor purity for several samples, which is expected for these longer times in culture (2weeks+). But how is this purity calculated? On the basis of the primary patient tumors or PDXs? It is unclear if the "original sample" here is the patient tumor or PDX, but, if not possible to report data for both, the more accurate comparison should be the input cells (thus PDX).**

In the original manuscript submission, we indicated that molecular comparisons were performed against the sequencing data from the original patient tumours obtained from ZERO childhood cancer precision medicine trial¹⁴ for both targeted sequencing and amplicon analysis.

We have now added the following consistent clarification:

original patient tumour (p.11)

- 9. Figure 4 is missing Ki67, PHOX2B, CD99 and SATB2 staining for the PDX samples of origin. Ideally, and if available, this figure should also include the original patient tumor staining for better comparison.**

Refer to response to Reviewer 1 Question 5.

- 10. These are PDXOs, so already selected for cases that favor growth in mice as a starting point. This initial bias should be acknowledged, as the results shown in Figure 5 are a good control and expected (which could be moved to the supplementary materials). Would be much more relevant and critical to test if bioprinting alters the engraftment capacity of primary clinical samples (which have not been grown as PDX to start).**

The PDX-derived cells used in this study are selected for their ability to grow in mice and served as a source of fresh patient-derived tumour cells. The primary aim of the tumorigenicity experiment (**Figure 5**) was to assess whether the 3D bioprinting process and subsequent expansion in hydrogels affect the tumorigenic potential of the cells. Our findings showed no phenotypic alterations following bioprinting and expansion. We believe this result is critical to understanding the impact of the bioprinting process on in vivo tumour behaviour and therefore should remain in the main text. A comparative analysis of growth rates between clinical samples not previously propagated in mice and those post-bioprinting falls outside the scope of this study.

- 11. *"Patient-derived cells were bioprinted, cultured between 7 to 14 days and subjected to an established ZERO pipeline for ex vivo drug screening using our customized library"***

Is there an SOP followed? Are samples of different tumor types grown for different time or is this varying patient-to-patient? This should be explained.

The ZERO ex vivo HTS drug screening pipeline approach has been well established^{2,15}. We provided relevant references before the abovementioned sentence in the original manuscript. The Methods section provides detailed protocols on high-throughput drug screening using 3D bioprinted samples that mimics ZERO pipeline in its overall approach.

We added the following clarification in the sentence:

“Patient-derived xenograft cells were bioprinted, cultured between 7 to 14 days (depending on the patient-specific variability in cell growth) and subjected to an established ZERO pipeline for ex vivo drug screening using our customised library”. (p.13, 308)

12. "Compared to other samples in the cohort, no patient-specific drug sensitivities in osteosarcoma samples were detected."

This differs from what was reported in Al Shihabi et al, is this due to the scale of the screening?

In our proof-of-concept study we designed a broad panel of 48 FDA approved drugs that targeted neuroblastoma, Ewing sarcoma and osteosarcomas (**Supplementary Table 6**). Al Shihabi ⁶ drug library consisted of 423 FDA-approved and experimental compounds with an average of 117 compounds per sample. The drug screening panels were individually tailored for each specific patient case with considerations to genetic alterations, histology and clinical background as well as an anticipated clinical treatment plan. Given this information, the comparison of ⁶ is irrelevant to our study, since the drug panel selection methodologies used in two studies are fundamentally different. Potential explanations for our observations were discussed in the original manuscript on p.13, 326-331.

13. "The hydrogels used in our study are highly tunable, well-defined and display reproducible stiffness and presentation of adhesive cues"

The hydrogels may be tunable, but the authors own data underpins how different compositions did not alter cell behavior of PDXOs. The limitations of Matrigel should be more accurately described - cultrex and Matrigel have been shown to work very well on sarcoma samples for instance, and the products are now very well QCed. In the absence of data included in the paper documenting differences, I suggest shortening this part of the discussion.

"The hydrogels used in our study are highly tunable, well-defined and display reproducible stiffness and presentation of adhesive cues" outlines the hydrogel properties used in our study, which aimed to identify formulations that support fresh tumour-derived material. We observed no significant differences in growth dynamics across varying stiffnesses or peptide compositions. Molecular features were preserved, and bioprinted samples closely resembled the original tumours. To address the reviewers concern we shortened the discussion from the previous manuscript (356-360), acknowledged the known limitations of Matrigel and other animal-derived ECMs⁷, and added the following sentence:

“Such properties could provide an advantage over the use of animal-derived ECM-like gels that have known limitations in replicating tissue-specific stiffness¹⁶ and variability associated with drug responses in cell culture-based experiments¹⁷.” (p.16, 389)

14. "In paediatric pre-clinical cancer research, a significant challenge is acquiring enough cellular material for direct drug screening, mainly due to the limited amount of initial tumour material available 8,13. Propagating cells through patient-derived xenografts has emerged as a particularly vital strategy for cultivating patient-derived cells from paediatric

tumours 25,68, however this approach can take months and is often unsuccessful. Within a shorter 7-14-day period, we successfully expanded patient- derived cells in ECM-mimic hydrogels in a high-throughput fashion-an achievement not always possible with conventional tissue culture."

This is confusing - according to their own Figure 5A, all samples used here are PDXOs, with delays in engraftment reported in Supplementary table 3 for 5 of the 8 cases. So the real timeline within this study is, say, 70 days +/- 7-14 days, not 7-14 days. Other studies have reported on generating and screening pediatric tumor organoids generated directly from clinical samples within this and shorter timelines - this is certainly feasible. But the discussion should actually reflect what is state of the art vs. what is reported within this article.

The reviewer is correct, our study focused on in situ expansion of cells using PDX models as a readily available source of patient-derived material for direct HTP drug screening. The 7–14 day timeframe demonstrates the feasibility of this approach in our proof-of-concept work.

15. "Our results confirm the importance of ex vivo HTP drug screening in support of clinical treatment decision making for paediatric patients and suggest that 3D bioprinted HTP drug screens could identify previously unrecognized chemotherapeutic and targeted drug vulnerabilities independent of tumour molecular profile."

That organoid-derived data should be supporting clinical decision is entirely dependent on clinical correlation. There is no clinical correlative data comparing the bioprinted PDXOs drug responses to patient outcomes within the paper to justify that, so the sentence should acknowledge this. Such correlations are available for non-bioprinted PDOs (see Peterziel et al, and Al Shihabi et al), but to my knowledge have not been systematically investigated for bioprinted PDOs or PDXOs.

We thank the reviewer for identifying the need to clarify the original sentence. It followed a discussion on observations that our 3D bioprinting approach could replicate preclinical drug testing outcomes, specifically noting that not all molecular vulnerabilities in patient-derived cells aligned with drug sensitivities identified in multiple paediatric ex vivo screens. To address the concern raised by the reviewer we have added the following sentence to the discussion:

"However, clinical correlation of drug responses from bioprinted patient-derived xenograft tumoroids to patient outcomes has not been systematically investigated in this study, unlike non-bioprinted patient-derived organoids where such correlations exist^{3,6}. Future studies and clinical trials will investigate the relationship between patient outcomes and preclinical drug responses in 3D bioprinted patient-derived models." (p. 18, 415)

Minor points:

16. "To create a tumour-like ECM for high-risk neuroblastoma and sarcoma, we selected hydrogels with two different stiffnesses levels - 1.1 kPa and 3 kPa - to closely mimic lung (~0.8 kPa) 41,42 and liver (~3 kPa) 43,44 tissues, respectively. These organs are metastatic sites for neuroblastoma⁴⁵ and sarcomas"

The manner in which these sentences are structured seem to suggest that neuroblastoma metastasize to the lung while sarcoma to liver, and can be confusing for non-experts. I suggest to reorganize for clarity.

We agree with the reviewer that this sentence requires reorganisation, and we have now modified for clarity:

*“To create a tumour-like extracellular matrix (ECM) for high-risk neuroblastoma and sarcoma, we selected hydrogels with two different stiffness levels: 1.1 kPa and 3 kPa. These stiffness levels closely mimic the lung (~0.8 kPa) and liver (~3 kPa) tissues, **which are frequent metastatic sites for sarcomas and neuroblastomas respectively.**”* p.7 (161)

17. Table S2 has a typo: Ewing Sarocoma

The typo (first column, second row) has been corrected to *Ewing Sarcoma*

18. *“Together our analysis demonstrates the feasibility and compatibility of 3D bioprinted matrix-embedded tumour organoids with a pre-clinical HTP drug screening pipeline as well as the ability to identify differential patient-specific responses and resistance using our workflow.”*

This sentence should clarify that the study is run on PDXOs. Similarly, the discussion should acknowledge this limitation.

We thank the reviewer for this observation. The following italicised and bolded text was added to clarify the source of the tumouroids, and the sentences were revised accordingly:

*“Together our analysis demonstrates the feasibility and compatibility of 3D bioprinted matrix-embedded **tumouroids from patient-derived xenografts** with a pre-clinical HTP drug screening pipeline as well as the ability to identify differential patient-specific drug responses using our workflow.”* (p.16, 269).

*“In this study, we established a 3D bioprinting workflow that enabled robust encapsulation of patient-derived tumour cells **from xenograft models** in the ECM-mimic hydrogels with defined mechanical and biochemical properties.”* (p.16, 380)

Furthermore, we added the following sentence to the discussion:

*“**Our study demonstrates feasibility of generating reproducible 3D ECM-mimic models for direct high-throughput drug screening bypassing conventional cell expansion approaches, aiming to preserve the molecular and phenotypic fidelity of patient-derived samples with potential clinical translatability. Future studies will link to a clinical trial to obtain additional patient samples to expand on this work.**”* (p.19, 441)

REFERENCES:

1. Lau, L.M.S., Khuong-Quang, D.A., Mayoh, C., Wong, M., Barahona, P., Ajuyah, P., Senapati, A., Nagabushan, S., Sherstyuk, A., Altekoester, A.K., et al. (2024). Precision-guided treatment in high-risk pediatric cancers. *Nat Med*. 10.1038/s41591-024-03044-0.
2. Mayoh, C., Mao, J., Xie, J., Tax, G., Chow, S.O., Cadiz, R., Pazaky, K., Barahona, P., Ajuyah, P., Trebilcock, P., et al. (2023). High-Throughput Drug Screening of Primary Tumor Cells Identifies Therapeutic Strategies for Treating Children with High-Risk Cancer. *Cancer Res*, OF1-OF17. 10.1158/0008-5472.CAN-22-3702.
3. Peterziel, H., Jamaladdin, N., ElHarouni, D., Gerloff, X.F., Herter, S., Fiesel, P., Berker, Y., Blattner-Johnson, M., Schramm, K., Jones, B.C., et al. (2022). Drug sensitivity profiling of 3D tumor tissue cultures in the pediatric precision oncology program INFORM. *NPJ Precis Oncol* 6, 94. 10.1038/s41698-022-00335-y.
4. Acanda De La Rocha, A.M., Berlow, N.E., Fader, M., Coats, E.R., Saghira, C., Espinal, P.S., Galano, J., Khatib, Z., Abdella, H., Maher, O.M., et al. (2024). Feasibility of functional precision medicine for guiding treatment of relapsed or refractory pediatric cancers. *Nat Med* 30, 990-1000. 10.1038/s41591-024-02848-4.
5. Ding, S., Hsu, C., Wang, Z., Natesh, N.R., Millen, R., Negrete, M., Giroux, N., Rivera, G.O., Dohlman, A., Bose, S., et al. (2022). Patient-derived micro-organospheres enable clinical precision oncology. *Cell Stem Cell* 29, 905-917 e906. 10.1016/j.stem.2022.04.006.
6. Al Shihabi, A., Tebon, P.J., Nguyen, H.T.L., Chantharasamee, J., Sartini, S., Davarifar, A., Jensen, A.Y., Diaz-Infante, M., Cox, H., Gonzalez, A.E., et al. (2024). The landscape of drug sensitivity and resistance in sarcoma. *Cell Stem Cell* 31, 1524-1542 e1524. 10.1016/j.stem.2024.08.010.
7. Aisenbrey, E.A., and Murphy, W.L. (2020). Synthetic alternatives to Matrigel. *Nat Rev Mater* 5, 539-551. 10.1038/s41578-020-0199-8.
8. Utama, R.H., Atapattu, L., O'Mahony, A.P., Fife, C.M., Baek, J., Allard, T., O'Mahony, K.J., Ribeiro, J.C.C., Gaus, K., Kavallaris, M., and Gooding, J.J. (2020). A 3D Bioprinter Specifically Designed for the High-Throughput Production of Matrix-Embedded Multicellular Spheroids. *iScience* 23, 101621. 10.1016/j.isci.2020.101621.
9. Utama, R.H., Tan, V.T.G., Tjandra, K.C., Sexton, A., Nguyen, D.H.T., O'Mahony, A.P., Du, E.Y., Tian, P., Ribeiro, J.C.C., Kavallaris, M., and Gooding, J.J. (2021). A Covalently Crosslinked Ink for Multimaterials Drop-on-Demand 3D Bioprinting of 3D Cell Cultures. *Macromol Biosci* 21, e2100125. 10.1002/mabi.202100125.
10. Thorel, L., Perreard, M., Florent, R., Divoux, J., Coffy, S., Vincent, A., Gaggioli, C., Guasch, G., Gidrol, X., Weiswald, L.B., and Poulain, L. (2024). Patient-derived tumor organoids: a new avenue for preclinical research and precision medicine in oncology. *Exp Mol Med* 56, 1531-1551. 10.1038/s12276-024-01272-5.
11. Abbas, Z.N., Al-Saffar, A.Z., Jasim, S.M., and Sulaiman, G.M. (2023). Comparative analysis between 2D and 3D colorectal cancer culture models for insights into cellular morphological and transcriptomic variations. *Sci Rep* 13, 18380. 10.1038/s41598-023-45144-w.
12. El Mokbel, N., Goyeneche, A.A., Prakash, R., Forgie, B.N., Abdalbari, F.H., Zeng, X., Tessier-Cloutier, B., Annie Leung, S.O., and Telleria, C.M. (2024). Comparison of two-dimensional and three-dimensional culture systems and their responses to chemotherapy in cells representing disease progression of high-grade serous ovarian cancer. *Biochem Biophys Rep* 40, 101838. 10.1016/j.bbrep.2024.101838.
13. Muguruma, M., Teraoka, S., Miyahara, K., Ueda, A., Asaoka, M., Okazaki, M., Kawate, T., Kuroda, M., Miyagi, Y., and Ishikawa, T. (2020). Differences in drug sensitivity between

- two-dimensional and three-dimensional culture systems in triple-negative breast cancer cell lines. *Biochem Biophys Res Commun* 533, 268-274. 10.1016/j.bbrc.2020.08.075.
14. Wong, M., Mayoh, C., Lau, L.M.S., Khuong-Quang, D.A., Pinese, M., Kumar, A., Barahona, P., Wilkie, E.E., Sullivan, P., Bowen-James, R., et al. (2020). Whole genome, transcriptome and methylome profiling enhances actionable target discovery in high-risk pediatric cancer. *Nat Med* 26, 1742-1753. 10.1038/s41591-020-1072-4.
 15. Lau, L.M.S., Mayoh, C., Xie, J., Barahona, P., MacKenzie, K.L., Wong, M., Kamili, A., Tsoli, M., Failes, T.W., Kumar, A., et al. (2022). In vitro and in vivo drug screens of tumor cells identify novel therapies for high-risk child cancer. *EMBO Mol Med* 14, e14608. 10.15252/emmm.202114608.
 16. Soofi, S.S., Last, J.A., Liliensiek, S.J., Nealey, P.F., and Murphy, C.J. (2009). The elastic modulus of Matrigel™ as determined by atomic force microscopy. *J Struct Biol* 167, 216-219. 10.1016/j.jsb.2009.05.005.
 17. Edmondson, R., Adcock, A.F., and Yang, L. (2016). Influence of Matrices on 3D-Cultured Prostate Cancer Cells' Drug Response and Expression of Drug-Action Associated Proteins. *PLoS One* 11, e0158116. 10.1371/journal.pone.0158116.

13th Aug 2025

Manuscript Number: MSB-2025-13287-T

Title: Engineered paediatric tumours retain patient tumour genotype and phenotype for precision medicine.

Author: MoonSun Jung

Valentina Poltavets

Joanna Skhinas

Gabor Tax

Alvin Kamili

Jinhan Xie

Sarah Ghamrawi

Philipp Graber

Jie Mao

Marie Wong

Louise Cui

Kathleen Kimpton

Pooja Venkat

Chelsea Mayoh

Angela Lin

Emmy Fleuren

Ashleigh Fordham

Zara Barger

John Grady

David Thomas

Eric Du

Nicole Graf

Mark Cowley

Andrew Gifford

Jamie Fletcher

Loretta Lau

M. Dolman

John Gooding

Maria Kavallaris

Dear Maria,

Thank you for submitting the revised version of your manuscript to Molecular Systems Biology. I have now reviewed the revised manuscript, point-by-point response, and the reviewer comments from EMBO Molecular Medicine.

Overall, we find that your study presents a compelling and relevant contribution to the field. We are pleased to inform you that we will be able to accept the manuscript for publication, pending a minor revision as outlined below.

Based on the reviewers' feedback, we believe that the technical concerns have been adequately addressed. In addressing the remaining concerns, we kindly ask that you provide a new point-by-point response specifically addressing the reviewers' comments regarding the clinically relevant advantages of your approach compared to existing methods as well as concerns about the level of technical innovation in writing. In the manuscript itself, we ask that you more clearly emphasize both the utility and the innovative aspects of your approach. Additionally, we suggest highlighting the "high-throughput" nature of your approach more explicitly in the manuscript title. No further experiments or analyses are required at this stage.

On a more editorial level, please address the following issues :

1. Please reduce the number of keywords to five and include them under a heading titled "Keywords".
2. The email address d.thomas@garvan.org.au bounced. Kindly provide a valid email address for this author.
3. Remove the "Author Contributions" section from the manuscript file.
4. Delete the "The Paper Explained" section, as it is not required for this journal.
5. Data Availability:

- Please remove the sections titled "Resource Availability", "Lead Contact", and "Materials Availability".
- Rename "Data and Code Availability" to "Data Availability". Include direct URLs for the S-BIAD2130 and EGAS00001008220 datasets, and ensure that both will be made publicly available upon acceptance of the manuscript.

6: Supplementary Figures and Tables

- Supplementary figures should be renamed to Figure EV1-EV5.
- Supplementary tables should be renamed to Table EV1-EV9.
- Please replace the "Supplementary Information" heading in the manuscript with "Expanded View Figure Legends."
- Remove the legends for the Expanded View tables from the manuscript text; these should appear only in the corresponding tables, each placed in a separate sheet.

7. Figure Callouts

- Please add the missing callout for Table S/EV8.
- There is a callout for Figure S88/EV8, but no such figure has been submitted. Please resolve this discrepancy.

8. Reference Formatting

- References must be formatted according to the Molecular Systems Biology style.
- List up to 10 co-authors before using et al. in the reference list.
- References should be listed in alphabetical order.
- DOIs should be included only for preprints or datasets. Please remove any DOIs that do not meet this criterion.

9. Please update the Author Checklist to reflect the correct journal name and manuscript number.

10. Please provide a "standfirst text" summarizing the study in one or two sentences (approximately 250 characters, including space), three to four "bullet points" highlighting the main findings and a "synopsis image" (550px width and 400-600 px height, PNG format) to highlight the paper on our homepage.

Here are a couple of examples:

<https://www.embopress.org/doi/10.15252/msb.20199356>

<https://www.embopress.org/doi/10.15252/msb.20209475>

<https://www.embopress.org/doi/10.15252/msb.209495>

11. Please address the following issues related to figure legends:

- Please note that the legends for figure S6 F is not provided in the sequential manner. This needs to be rectified.
- Please note that the exact p values are not provided in the legends of figures 1C, D; 2A-C; S3A-C; S5 A-C.

Click on the link below to submit your revised paper.

Kind regards,
Jingyi

Jingyi Hou, PhD
Senior Editor
Molecular Systems Biology

*** PLEASE NOTE *** As part of the EMBO Press transparent editorial process initiative (see our Editorial at <https://dx.doi.org/10.1038/msb.2010.72> , Molecular Systems Biology will publish online a Review Process File to accompany accepted manuscripts. When preparing your letter of response, please be aware that in the event of acceptance, your cover letter/point-by-point document will be included as part of this File, which will be available to the scientific community. More information about this initiative is available in our Instructions to Authors. If you have any questions about this initiative, please contact the editorial office (msb@embo.org).

Response to Editor comments:

- *Provide a new point-by-point response specifically addressing the reviewers' comments regarding the clinically relevant advantages of your approach compared to existing methods as well as concerns about the level of technical innovation in writing.*

Author: Addressed below under "Response to Reviewer"

- *In the manuscript itself, we ask that you more clearly emphasise both the utility and the innovative aspects of your approach.*

Author: We appreciate this suggestion and have tried to be clearer on the utility and innovative aspects of our approach. This is particularly evident in the discussion.

- *Additionally, we suggest highlighting the "high-throughput" nature of your approach more explicitly in the manuscript title. No further experiments or analyses are required at this stage.*

Author: The HTP nature of our approach has been more explicitly outlined in the title, and in the manuscript.

- *Please reduce the number of keywords to five and include them under a heading titled "Keywords".*

Author: Corrected

- *The email address d.thomas@garvan.org.au bounced. Kindly provide a valid email address for this author.*

Author: Corrected

- Editorial Changes – all have been completed as requested. The data will be made available once the manuscript is accepted for URLs S-BIAD2130 and EGAS00001008220 datasets,.
- Regarding exact p-values, we calculated correlations and p-values in R and GraphPad Prism. Both programs only report up to 16 significant digits, so extremely small p-values can't be shown exactly and the software displays them as below its limit (e.g., "< ..."). That's a software constraint rather than our reporting choice. Accordingly, we report p-values in that form and focus on the effect size (the correlation) as the more informative result.

Response to Reviewer #1: Comments on Novelty/Model System**Reviewer Comment:**

"Although this study demonstrated that the proposed platform is compatible with pediatric xenograft tissue and enables high-throughput screening (HTS), the additional data provided for primary tumor tissues raised some concerns. Specifically, the growth of the primary tumors was quite slow—one started growing on Day 14 and another on Day 20. From a

modeling perspective, it is unclear whether this platform offers a significant advantage over conventional organoid methods or meaningfully addresses the key challenges associated with modeling rare cancers. Furthermore, from a technical standpoint, the platform does not appear to introduce substantial innovation, as similar devices have been reported in previous studies.”

Author Response:

We appreciate the reviewer’s thoughtful comments and would like to clarify the novelty and advantages of our platform:

1. Growth of Primary Tumours:

The slower growth observed in direct patient samples (e.g., zccs486) reflects the inherent biological variability and challenges of culturing rare paediatric cancers ex vivo. Importantly, our platform successfully supported expansion of these samples in defined ECM-mimic hydrogels, which is a significant achievement given the known difficulty of growing such tumours directly from biopsy material (see Discussion). This contrasts with conventional organoid methods that often require in vivo expansion or fail to support growth of mesenchymal tumours like osteosarcoma.

2. Advantages Over Conventional Organoids:

- Our 3D bioprinting approach incorporates defined ECM-mimicking peptides and tunable stiffness, which better replicate the tumour microenvironment compared to Matrigel-based organoids that suffer from batch variability and undefined composition (Introduction, lines 99-102).
- The platform enables standardised, scalable, and reproducible bioprinting of tumouroids in minutes, which is not feasible with manually seeded organoids (Discussion).
- We demonstrated retention of genotype, phenotype, and tumorigenicity post-bioprinting, including preservation of structural variants and disease-specific markers (Results, Fig. 3–5), which has not been well characterised in conventional organoid systems.
- Although not heavily emphasised in the paper, the ability to expand the patient-derived cells, extract them viably from the hydrogel, cryopreserve them, and at a later date, thaw and use them for tumourigenicity or other studies (Fig 5), highlights the potential to expand and biobank rare tumour samples using our approach.

3. Innovation and Technical Advancement:

- This is the first study to apply high-throughput 3D bioprinting using ECM-mimic hydrogels for paediatric patient-derived and primary cells in a drug screening pipeline. This is important as embryonal tumours have a low mutational burden and are driven by a few key genetic events (e.g., MYCN amplification in neuroblastoma, RB1 loss in retinoblastoma). These events may not be sufficient to induce transformation in vitro unless the cells are in a very specific developmental state making them difficult to grow and characterise ex vivo (reviewed in Custers et al. *Frontiers in Cell & Developmental Biology* 9:640633, 2021). Adult carcinomas, on the other hand, accumulate mutations over time, which can make them more robust and easier to propagate in culture. (Sentences 2-3 of this paragraph have been added to paragraph 3 of the discussion).

- Unlike prior models, our platform allows direct in situ drug screening without requiring prior expansion, reducing time and cost (Results, Fig. 6).
- The bioprinting system used (RASTRUM) offers precise control over cell distribution, hydrogel composition, and mechanical properties, which is a substantial technical advancement over existing 3D culture methods.

Response to Reviewer #2: Remarks for Author

Reviewer Comment:

“The authors have addressed my questions 1 and 3. However, regarding question 2, I still do not see a clear technological or clinical advancement in the use of 3D bioprinted organoids. While these findings suggest consistency between models, they do not demonstrate a distinct advantage of 3D bioprinted tumouroids over established models. In fact, the process of generating 3D bioprinted tumouroids/organoids from PDX is time-consuming and costly. Compared to patient-derived tumouroids or even 2D/3D cultures of PDX, the added value—whether technological or clinical—remains unclear based on the current evidence.”

Author Response:

We thank the reviewer for their continued engagement and would like to highlight the distinct advantages of our 3D bioprinted tumouroid platform:

1. Technological Advancement:

- Our platform enables rapid and reproducible HTP bioprinting of tumouroids using ECM-mimic hydrogels with defined mechanical and biochemical properties, overcoming limitations of Matrigel and other animal-derived matrices (Discussion, paragraph 2).
- The bioprinting process is automated and scalable, allowing high-throughput production of tumouroids in standard formats (e.g., 384-well plates for HTP drug screening), which is not feasible with manual organoid culture (Methods: Bioprinting section).
- We demonstrated preservation of molecular fidelity, including STR/SNP matching, RNAseq correlation ($r > 0.96$), and retention of tumourigenic capacity (Results, Fig. 3–5, EV3), validating the robustness of the bioprinted models.

2. Clinical Relevance:

- The platform supports direct bioprinting of patient samples, including rare and difficult to grow tumours, enabling drug screening without prior expansion (Results, Fig. EV6).
- We identified patient-specific drug sensitivities (e.g., venetoclax and alisertib in MYCN-amplified neuroblastoma) that were not predicted by molecular profiling alone, highlighting the value of functional testing (Results, Fig. 6D–F).
- Compared to conventional organoids, our platform offers faster turnaround (7–14 days for patient-derived xenograft cells and 14–20 days for primary tumour cells vs. months for PDX expansion), which is critical for clinical decision-making in high-risk paediatric cancers.

3. Cost and Efficiency:

- While initial setup (purchase of printer) may be resource-intensive, the standardisation and scalability of our workflow reduce long-term costs and

variability. Moreover, the ability to bypass in vivo expansion and conduct in situ drug screening offers significant time, resource savings (Discussion, paragraph 4) and reduces the use of animals in research.

We hope these clarifications address the reviewer's concerns and underscore the novelty, utility, and translational potential of our 3D bioprinted tumoroid platform.

9th Sep 2025

Manuscript number: MSB-2025-13287R

Title: High-throughput 3D engineered paediatric tumour models for precision medicine.

Dear Dr. Kavallaris,

Thank you again for sending us your revised manuscript. We are now satisfied with the modifications made and I am pleased to inform you that your paper has been accepted for publication.

Sincerely,
Jingyi

Jingyi Hou, PhD
Senior Editor
Molecular Systems Biology
